# On Effective Scheduling of Model-based Reinforcement Learning

**Hang Lai**[* 1], **Jian Shen**[* 1], **Weinan Zhang**[† 1], **Yimin Huang**[2],
**Xing Zhang**[2], **Ruiming Tang**[2], **Yong Yu**[1], **Zhenguo Li**[2]
[1]Shanghai Jiao Tong University, [2]Huawei Noah's Ark Lab
{laihang, wnzhang}@apex.sjtu.edu.cn

## Abstract

Model-based reinforcement learning has attracted wide attention due to its superior sample efficiency. Despite its impressive success so far, it is still unclear how to appropriately schedule the important hyperparameters to achieve adequate performance, such as the real data ratio for policy optimization in Dyna-style model-based algorithms. In this paper, we first theoretically analyze the role of real data in policy training, which suggests that gradually increasing the ratio of real data yields better performance. Inspired by the analysis, we propose a framework named AutoMBPO to automatically schedule the real data ratio as well as other hyperparameters in training model-based policy optimization (MBPO) algorithm, a representative running case of model-based methods. On several continuous control tasks, the MBPO instance trained with hyperparameters scheduled by AutoMBPO can significantly surpass the original one, and the real data ratio schedule found by AutoMBPO shows consistency with our theoretical analysis.

## 1  Introduction

Deep model-free reinforcement learning (MFRL) has achieved great successes in complex decision-making problems such as Go [26] and robotic control [10], to name a few. Although MFRL can achieve high asymptotic performance, a tremendous number of samples collected in interactions with the environment are required. In contrast, model-based reinforcement learning (MBRL), which alternately learns a model of the environment and derives an agent with the help of current model estimation, is considered to be more sample efficient than MFRL [27].

MBRL methods generally fall into different categories according to the specific usage of the learned model [25, 33]. Among them, Dyna-style algorithms [28] adopt some off-the-shelf MFRL methods to train a policy with both real data from environment and imaginary data generated by the model. Since Dyna-style algorithms can seamlessly take advantage of innovations in MFRL literature and have recently shown impressive performance [12, 30], this paper mainly focuses on Dyna-style algorithms. Although some theoretical analysis [17, 12, 22] and thorough empirical evaluation [30] have been conducted in the previous literature, it remains unclear how to appropriately schedule the hyperparameters to achieve optimum performance when training a Dyna-style MBRL algorithm. In practice, many important hyperparameters may primarily affect performance.

Firstly, since Dyna-style MBRL algorithms consist of model learning and policy optimization, how to balance the alternate optimization of these two parts is a crucial problem [22]. Intuitively, insufficient model training may lead to inaccurate estimation while excessive model training will cause overfitting, and a small amount of policy training may not utilize the model adequately while exhaustive policy training will exploit the model's deficiencies. Moreover, the rollout length of the imaginary trajectory

---

[*]equal contribution. [†]Weinan Zhang is the corresponding author.

35th Conference on Neural Information Processing Systems (NeurIPS 2021).

is also critical [12] since too short rollout length fails to sufficiently leverage the model to plan forward, while too long rollout length may bring disastrous compounding error [1]. Janner et al. [12] manually design a schedule that linearly increases the rollout length across epochs. Finally, when using both real samples and imaginary samples to train the policy, how to control the ratio of the two datasets remains unclear. Janner et al. [12] fix the ratio of real data as $5\%$, while Kalweit and Boedecker [13] use the uncertainty to adaptively choose the ratio, which tends to use more imaginary samples initially and gradually use more real samples afterward. According to these existing works, the optimal hyperparameters in model-based methods may be dynamic during the whole training process, which further strengthens the burden of manually scheduling the hyperparameters.

Based on these considerations, in this work, we aim to investigate how to appropriately schedule these hyperparameters, i.e., real data ratio, model training frequency, policy training iteration, and rollout length, to achieve optimal performance of Dyna-style MBRL algorithms. Although real data ratio is an essential factor empirically [13, 12], it has not yet been studied thoroughly in theory. To bridge this gap, we first derive a return discrepancy upper bound for model-based value iteration, which reveals that *gradually increasing the ratio of real data yields a tighter bound than choosing a fixed value*. Inspired by the analysis, considering the complex interplay between real data ratio and other hyperparameters in practice, we develop AutoMBPO to automatically determine the joint schedule of the above hyperparameters in model-based policy optimization (MBPO), a representative running case of Dyna-style MBRL algorithms. Specifically, AutoMBPO introduces a parametric hyper-controller to sequentially choose the value of hyperparameters in the whole optimization process to maximize the performance of MBPO. We apply AutoMBPO to several continuous control tasks. On all these tasks, the MBPO instance trained with hyperparameters scheduled by AutoMBPO can significantly surpass the one with original configuration [12]. Furthermore, the hyperparameter schedule found by AutoMBPO is consistent with our theoretical analysis, which yields a better understanding of MBPO and provides insights for the design of other MBRL methods.

## 2 Related Work

Model-based reinforcement learning (MBRL) has been widely studied due to its high sample efficiency. MBRL methods can be roughly categorized into four types according to different model usage: (i) Dyna-style algorithms [28, 17, 4, 14, 12, 15] leverage the model to generate imaginary data and adopt some off-the-shelf MFRL algorithms to train a policy using both real data and imaginary data; (ii) shooting algorithms [19, 3] use model predictive control (MPC) to plan directly without explicit policy; (iii) analytic-gradient algorithms [7, 16, 5] search policies with back-propagation through time by exploiting the model derivatives; (iv) model-augmented value expansion algorithms [9, 2] utilize model rollouts to improve the target value for temporal difference (TD) updates. This paper mainly focuses on the first category.

Dyna-style algorithms [28] typically alternate between sampling real data, learning dynamics models, generating imaginary data, and optimizing policy. Many efforts have been made to improve one or more of these steps. For model learning, the deep ensemble technique [21, 3, 14] has been leveraged to resist overfitting. As for imaginary data generation, it is suggested to generate short rollouts to reduce compounding model error [12], and the model's uncertainty is further incorporated to choose reliable imaginary data dynamically [20]. When optimizing the policy, we can use merely imaginary data [17, 14] or a mixture of real data and imaginary one in a fixed ratio [12]. Moreover, the ratio can also be dynamically adjusted according to the estimated uncertainty of the Q-function [13]. Besides, Rajeswaran et al. [22] present a game-theoretic framework for MBRL by formulating the optimization of model and policy as a two-player game.

As for automatic hyperparameter optimization for MBRL, Zhang et al. [32] utilize population based training (PBT) to tune the hyperparameters of a shooting algorithm dynamically. The reinforcement on reinforcement (RoR) framework proposed by Dong et al. [8] applies an RL algorithm to control the sampling and training process of an MBRL method. Our work differs from theirs in four major aspects: (1) We provide a theoretical analysis for the hyperparameter being scheduled, which is not included in their work. (2) In terms of problem settings, RoR assumes one can directly access an initial state without further interaction from that state, which is impractical for most RL problems. (3) As for the method design, our formulated hyper-MDP in Section 5.1 is totally different from RoR's, including state, action and reward. In detail, our state definition contains more training information, and the reward design is based on the return rather than a single-step reward in the target-MDP. (4)

Finally, in the experiments in Section 6, our method achieves better performance and generalization in all environments, and the empirical findings also show differences.

## 3 Preliminaries

We first briefly introduce the RL problem with notations used throughout this paper and the previous MBRL method on which our proposed framework is based.

### 3.1 Reinforcement Learning

In reinforcement learning (RL), there originally exists a Markov decision process (MDP), which we will call *target-MDP* throughout the paper. The target-MDP is defined by the tuple $(\mathcal{S}, \mathcal{A}, T, r, \gamma)$. $\mathcal{S}$ and $\mathcal{A}$ are the state and action spaces, respectively, and $T(s' \mid s, a)$ is the transition density of state $s'$ given action $a$ taken under state $s$. The reward function is denoted as $r(s, a)$, and $\gamma \in (0, 1)$ is the discount factor. The goal of RL is to find the optimal policy $\pi^*$ that maximizes the expected return, denoted by $\eta$:

$$\pi^* := \arg\max_\pi \eta[\pi] = \arg\max_\pi \mathbb{E}_\pi \Big[ \sum_{t=0}^\infty \gamma^t r(s_t, a_t) \Big], \tag{1}$$

where $s_{t+1} \sim T(s \mid s_t, a_t)$ and $a_t \sim \pi(a \mid s_t)$. Generally, the ground truth transition $T$ is unknown, and MBRL methods aim to construct a model $\hat{T}$ of the transition dynamics to help improve the policy.

To evaluate a reinforcement learning method (no matter whether it is model-free or model-based), we mainly care about two metrics: (1) asymptotic performance: the expected return of the converged policy in real environments; (2) sample efficiency: the number of samples $(s, a, s', r)$ collected in target-MDP to achieve some performance. Therefore, an effective RL method means it achieves higher expected returns with fewer real samples.

### 3.2 Model-Based Policy Optimization

Model-based policy optimization (MBPO) [12] is a state-of-the-art MBRL method which will be used as the running case in our framework. MBPO learns a bootstrapped ensemble of probabilistic dynamics models $\{\hat{T}_\theta^1, ..., \hat{T}_\theta^E\}$ where $E$ is the ensemble size. Each individual dynamics model in the ensemble is a probabilistic neural network that outputs a Gaussian distribution with diagonal covariance $\hat{T}_\theta^i (s'|s, a) = \mathcal{N} \left( \mu_\theta^i (s, a), \Sigma_\theta^i (s, a) \right)$. These models are trained independently via maximum likelihood with different initializations and training batches. The corresponding loss function is

$$\mathcal{L}_{\hat{T}}(\theta) = \sum_{n=1}^N [\mu_\theta (s_n, a_n) - s_{n+1}]^\top \Sigma_\theta^{-1} (s_n, a_n) [\mu_\theta (s_n, a_n) - s_{n+1}] + \log \det \Sigma_\theta (s_n, a_n).$$

In practical implementation, an early stopping trick is adopted in training the model ensemble. To be more specific, when training each individual model, a hold-out dataset will be created, and the training will early stop if the loss evaluated on the hold-out data does not decrease.

Every time the ensemble models are trained, they are then used to generate $k$-length imaginary rollouts, which begin from states sampled from real data and follow the current policy. Each time the policy interacts with the target-MDP, it will be trained for $G$ gradient updates using both real data and imaginary data in a certain ratio. To make it clear, in each batch for policy training, we denote the proportion of real data to the whole batch as $\beta$ and the proportion of imaginary data as $1 - \beta$. We call $\beta$ the ***real ratio*** and call $G$ the ***policy training iteration*** throughout the paper. The policy optimization algorithm is SAC (Soft Actor-Critic) [11], and the loss functions for the actor $\pi_\omega$ and the critic $Q_\phi$ are

$$\mathcal{L}_\pi(\omega) = \mathbb{E}_{s_t \sim D} \Big[ \mathbb{E}_{a_t \sim \pi_\omega} [\alpha \log(\pi_\omega(a_t|s_t)) - Q_\phi(s_t, a_t)] \Big], \tag{2}$$

$$\mathcal{L}_Q(\phi) = \mathbb{E}_{(s_t, a_t) \sim D} \Big[ \frac{1}{2} \big( Q_\phi(s_t, a_t) - (r(s_t, a_t) \tag{3}$$
$$+ \gamma \mathbb{E}_{s_{t+1}, a_{t+1}} [Q_{\hat{\phi}}(s_{t+1}, a_{t+1}) - \alpha \log \pi_\omega(a_{t+1}|s_{t+1})]) \big)^2 \Big],$$

where $Q_{\hat{\phi}}$ is the target Q-function.

# 4 Analysis of real ratio schedule

Previous theoretical findings in MBRL mainly focus on the optimization of model and policy [17, 22] or the rollout length [12, 15], while the usage of real data in policy optimization remains unclear. Kalweit and Boedecker [13] only use real data near convergence but accept noisier imaginary data initially, which seems confusing since the model may not be accurate enough at the beginning. In this section, we provide a theoretical analysis of the real ratio schedule by deriving a return discrepancy upper bound of sampling-based Fitted Value Iteration (FVI) in model-based settings. The analysis mainly follows Munos and Szepesvári [18], where all the training data are sampled from the underlying dynamics. We extend the framework to a more general case: the data can either be sampled from the underlying dynamics w.p. (with probability) $\beta$ or from a learned model w.p. $1 - \beta$, which we call $\beta$-*mixture sampling-based FVI*. Besides, we consider deterministic environments here, i.e., given $(s, a)$, the next state $s' = T(s, a)$ is unique, just as in all the environments in Section 6.

For the sake of specificity, we first present a detailed description of the $\beta$-mixture sampling-based FVI algorithm. Let $\mathcal{F}$ be the value function space, and $V_0 \in \mathcal{F}$ be the initial value function. The FVI algorithm produces a sequence of functions $\{V_k\}_{0 \le k \le K} \subset \mathcal{F}$ iteratively. Given $V_k$ and $N$ sampled states $\{s_1, ..., s_N\}$, the value function of the next iteration $V_{k+1}$ is computed as

$$\hat{V}_k(s_i) = \max_{a \in \mathcal{A}} \left( r_i + \gamma V_k(s_i') \right), i = 1, 2, \ldots, N, \tag{4}$$

$$V_{k+1} = \operatorname*{argmin}_{f \in \mathcal{F}} \sum_{i=1}^{N} \left| f(s_i) - \hat{V}_k(s_i) \right|^p, \tag{5}$$

where $s_i$ is uniformly sampled from a certain state distribution $\mu$, and for each possible action $a$, the next state $s_i'$ is sampled from the underlying dynamics w.p. $\beta$ or from the learned model w.p. $1 - \beta$. As final preparatory steps, we define the *Bellman operator* for deterministic MDPs $B : \mathcal{F} \to \mathcal{F}$ as

$$BV(s) = \max_{a \in \mathcal{A}} \left\{ r(s, a) + \gamma V(s') \right\},$$

where $s' = T(s, a)$; and for function $g$ over $\mathcal{X}$, $\|g\|_{p,\mu}$ is defined as $\|g\|_{p,\mu}^p = \int |g(x)|^p \mu(dx)$. Following Munos and Szepesvári [18], we begin with the error bound in a single iteration.

**Lemma 4.1.** *(Single Iteration Error Bound) Let $V_k$ and $V_{k+1}$ be the value functions of iteration $k$ and $k+1$, and $V_{max} = r_{max}/(1-\gamma)$. For $p \ge 1$ and a certain state distribution $\mu$, let the inherent Bellman error of the value function space $\mathcal{F}$ be defined by $d_{p,\mu}(B\mathcal{F}, \mathcal{F}) = \sup_{V \in \mathcal{F}} \inf_{f \in \mathcal{F}} \|f - BV\|_{p,\mu}$. Assume the value functions are $K_V$-Lipschitz continuous, i.e., for any $k$ and states pair $(s_i, s_j)$, it holds that $|V_k(s_i) - V_k(s_j)| \le K_V \|s_i - s_j\|_2$. And assume the $L^2$-norm model error between the real next state $s'$ and the predicted ones $\hat{s}'$ obeys a half-normal distribution, i.e., the probability density function $f(\|\hat{s}' - s'\|_2) = \frac{2}{\sqrt{2\pi}\sigma} \exp\left(-\frac{\|\hat{s}' - s'\|_2^2}{2\sigma^2}\right)$. Then for any $\epsilon_0, \delta_0 > 0$, the inequality*

$$\|V_{k+1} - BV_k\|_{p,\mu} \le d_{p,\mu}(B\mathcal{F}, \mathcal{F}) + \epsilon_0$$

*holds w.p. at least $1 - \delta_0$ provided that*

$$N > 128 \left(\frac{8V_{\max}}{\epsilon_0}\right)^{2p} \left( \log(1/\delta_0) + \log(32N) \right), \tag{6}$$

*and*

$$\sigma < \frac{\epsilon_0}{4\gamma K_V \Phi^{-1}\left(1 - \delta_0/\left(8N|\mathcal{A}|(1 - \beta)\right)\right)}. \tag{7}$$

*Proof.* See *Appendix A*, *Lemma A.2.* □

According to the model error assumption (more details in Appendix B), the model will give precise predictions if $\sigma$ is small enough. Therefore, Lemma 4.1 states that with an accurate model and sufficient large $N$, the errors between $BV_k$ and $V_{k+1}$ in each iteration can be bounded with high probability. Now let us turn to bound the return discrepancy between the greedy policy w.r.t. $V_K$ and the optimal policy.

**Theorem 4.1.** *($\beta$-mixture sampling-based FVI bound) Under the concentrability assumption and the same assumptions of Lemma 4.1, let $\rho$ be an arbitrary state distribution, $C_{\rho,\mu}$ be the discounted-average concentrability coefficient and $\pi_K$ be the greedy policy w.r.t. $V_K$. Let $V^{\pi_K}$ and $V^*$ be the expected return of executing $\pi_K$ and the optimal policy $\pi^*$ in real environment, respectively. Define $N_{\text{real}} = N \cdot |\mathcal{A}| \cdot \beta$ as the expected number of real samples. Then the following bound holds w.p. at least $1 - \delta$:*

$$
\|V^* - V^{\pi_K}\|_{p,\rho} \leq \frac{2\gamma}{(1-\gamma)^2} C_{\rho,\mu}^{1/p} d_{p,\mu}(B\mathcal{F}, \mathcal{F}) + O\left(\left(\frac{\beta|\mathcal{A}|}{N_{\text{real}}}\left(\log\left(\frac{N_{\text{real}}}{\beta|\mathcal{A}|}\right) + \log\left(\frac{K}{\delta}\right)\right)\right)^{\frac{1}{2p}}\right)
$$
$$
+ O\left(\Phi^{-1}\left(1 - \frac{\beta\delta}{8KN_{\text{real}}(1-\beta)}\right)\sigma\right) + O\left(\gamma^{K/p}V_{\max}\right). \tag{8}
$$

*Proof.* See *Appendix A, Theorem A.1.* $\qquad\square$

**Remark.** The definition of the concentrability assumption is provided in Appendix A. Theorem 4.1 gives an upper bound on the return discrepancy. To achieve small return discrepancy, we hope to minimize the upper bound. For the first term in the bound, the approximation error $d_{p,\mu}(B\mathcal{F}, \mathcal{F})$ can be made small by selecting $\mathcal{F}$ to be large enough, and the last term arises due to the finite number of iterations. For the remaining two terms, as $\beta/N_{\text{real}}$ increases, the second term increases while the third term decreases. Therefore, there exists an optimal value for $\beta/N_{\text{real}}$ to trade off these two terms. In online MBRL, $N_{\text{real}}$ increases as real samples are continuously collected from the environment throughout the training process, so that gradually increasing the real ratio $\beta$ is promising to achieve good performance according to the upper bound.

From another perspective, the second term corresponds to the estimation error of using limited data to learn the value functions. By utilizing a learned model, we can generate as much data as we want, which, however, tends to suffer from the model error represented by the third term. The hyperparameter $\beta$ serves as a trade-off between these two kinds of errors. Intuitively, with more real data available, the pressure of limited data is alleviated, which means we can increase $\beta$ accordingly to reduce the reliance on inaccurate models.

## 5 The AutoMBPO Framework

The analysis in the previous section shows that the optimal real ratio $\beta$ during training should gradually increase according to the current optimization states, such as the number of real samples and model error. These states may also be affected by other hyperparameters, e.g., model training frequency and rollout length, which makes the hyperparameter scheduling non-trivial in practice. In order to investigate the optimal joint scheduling of these hyperparameters, we propose a practical framework named AutoMBPO to schedule these hyperparameters automatically when training an MBRL algorithm. The high-level idea of AutoMBPO is to formulate hyperparameter scheduling as a sequential decision-making problem and then adopt some RL algorithm to solve it. Here we choose MBPO [12] as a representative running case since MBPO has achieved remarkable success in continuous control benchmark tasks and can naturally take all the hyperparameters we care about into consideration (a comparison of the hyperparameters that can be scheduled in different MBRL algorithms is provided in Appendix C). We illustrate the overview framework of AutoMBPO in Figure 1 and detail the algorithm in Algorithm 1.

### 5.1 Hyper-MDP Formulation

We formulate the problem of determining the hyperparameter schedule as a new sequential decision-making problem, which we will call *hyper-MDP*. In the hyper-MDP, the hyper-controller needs to decide the hyperparameter values throughout one complete run of the MBPO learning process. Below we elaborate on several critical aspects of the hyper-MDP formulation used in this paper.

**Episode.** An episode of the hyper-MDP consists of the whole process of training an MBPO instance from scratch, which contains multiple episodes of the target-MDP. We will terminate an episode of the hyper-MDP if $N$ samples have been collected in the target-MDP. To be more specific, if the episode length of the target-MDP is $H$, and we only train MBPO for $m$ episodes, then we have $N = H \cdot m$.

**State.** The state should be defined to include all the information of MBPO training to help hyper-controller make decisions accordingly. We use a concatenation of several features to represent the state. As suggested by Theorem 4.1, the state should contain $N_{\text{real}}$, the number of real samples so far, and the model error, which can be estimated by the model loss $\mathcal{L}_{\hat{T}}$ on a validation set. Besides, we also add other features to better capture the current optimization situation: (1) critic loss: $\mathcal{L}_Q$, which to some extent represents how well the value function fits the training data; (2) policy change: $\epsilon_\pi = E_{(s,a)\sim\pi_D}[||\pi(a|s)-\pi_D(a|s)||]$, which evaluates the distance of current policy $\pi$ and the data-collecting policy $\pi_D$ on sampled data. In prac-

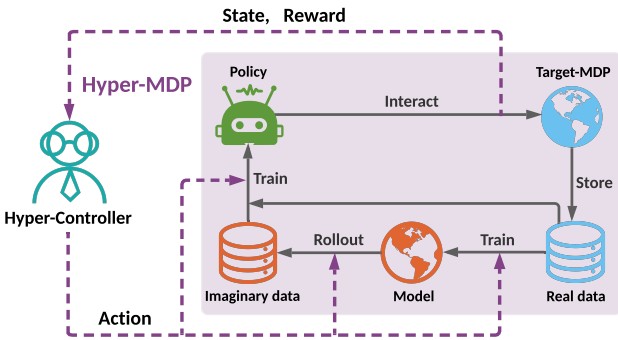

Figure 1: Illustration of AutoMBPO framework. The hyper-controller (left) treats the whole learning process of MBPO (right) as its hyper-MDP and takes action according to the state containing all the information of MBPO training. The action is to adjust the hyperparameters of MBPO, e.g., real ratio, model training frequency, policy training iteration, and rollout length. Afterward, a reward is given to the hyper-controller based on the performance of MBPO policy in target-MDP.

tice, we calculate $\epsilon_\pi$ using recently collected data to reflect the updating speed of policy roughly; (3) validation metrics: we use the average return on the target-MDP to represent the performance of the policy; (4) current hyperparameters: the hyperparameter values used currently by MBPO. All these features are normalized into $[0,1]$ to make the learning process stable.

**Action.** The action set in the hyper-MDP should contain all the hyperparameters we care about. Firstly, we would like to adjust the ratio of real data when optimizing the policy. Secondly, we also consider investigating the model training and policy learning frequency to balance their alternate optimization [22]. Lastly, the rollout length when generating imaginary rollouts is crucial in MBPO, but it remains unclear how to choose its value [12]. Besides, since there exists an optimal value for $\beta/N_{\text{real}}$ according to the theoretical analysis and $N_{\text{real}}$ gradually increases during training in practice, for the action space, adjusting the real ratio by multiplying a constant is enough to get an effective schedule. To sum up, the action set and the corresponding space of each action are: (1) adjusting the real ratio $\beta$: $\{\times c^{-1}, \times 1, \times c\}$, where c is a constant larger than 1; (2) deciding whether to train the model: $\{0,1\}$. Note that in MBPO, the model training iterations are implicitly decided by an early stopping trick, so we only decide whether to train the model; (3) adjusting the policy training iteration $G$: $\{-1,0,+1\}$; (4) adjusting the model rollout length $k$: $\{-1,0,+1\}$. In practice, the actions are taken every $\tau$ real samples are collected where $\tau$ is the hyperparameter of the hyper-controller.

**Reward.** One direct way is to define a reward at the end of an episode in hyper-MDP, which, however, will lead to the sparse reward problem. We define our reward as follows: every $H$ real samples, i.e., one episode in target-MDP, we evaluate the MBPO policy on target-MDP and define the reward as the average evaluated return.

### 5.2 Hyper-Controller Learning

After formulating the hyper-MDP, we are now ready to learn a hyper-controller in the hyper-MDP. In fact, we can use any off-the-shelf reinforcement learning algorithm, and we choose Proximal Policy Optimization (PPO) [24] as the hyper-controller algorithm since PPO has achieved considerable success in the optimization schedule problem [31]. To be more specific, we maximize the following clipped PPO objective:

$$\mathcal{J}(\Omega) = \hat{\mathbb{E}}_t\left[\min\left(r_t(\Omega)\hat{A}_t, \text{clip}\left(r_t(\Omega), 1-\epsilon, 1+\epsilon\right)\hat{A}_t\right)\right],$$

where $r_t(\Omega) = \mathbf{MC}_\Omega(a_t|s_t)/\mathbf{MC}_{\Omega_{\text{old}}}(a_t|s_t)$ and $\mathbf{MC}_\Omega$ is the hyper-controller policy parameterized by $\Omega$.

There are multiple choices of the advantage function [23], and we use the baseline version of the Monte-Carlo returns to reduce the variance:

$$\hat{A}_t = \Sigma_{i=t}^{m\cdot H/\tau}(R_i - R'_i). \tag{9}$$

---

**Algorithm 1:** AutoMBPO

---

1   Initialize a hyper-controller **MC**
2   **repeat**
3      Initialize a MBPO instance; initialize the hyper-controller sample count $i = 0$
4      **for** $m$ *target-MDP episodes* **do**
5          **for** $h = 0 : H - 1$ *real timesteps (an episode in target-MDP)* **do**
6              Interact with the target-MDP using the MBPO policy; add the real samples to $\mathcal{D}_{\text{env}}$
7              **if** $h \bmod \tau = 0$ **then**
8                  Calculate the current state $\mathbf{S}_i$ for hyper-MDP; take an action based on $\mathbf{S}_i$:
                     $\mathbf{A}_i \leftarrow \mathbf{MC}(\mathbf{S}_i)$
9                  Decide whether to train the model, and adjust the hyperparameters $k, G, \beta$
                     according to $\mathbf{A}_i$
10              **for** $F$ *model rollouts* **do**
11                  Sample $s_t$ uniformly from $\mathcal{D}_{\text{env}}$; perform $k$-step rollout from $s_t$; add the
                     imaginary data to $\mathcal{D}_{\text{model}}$
12              **for** $G$ *gradient updates* **do**
13                  Train the MBPO policy using SAC on $\mathcal{D}_{\text{env}}$ and $\mathcal{D}_{\text{model}}$ in the ratio $\beta$
14              **if** $h = H - 1$ **then**
15                  Evaluate MBPO and construct the reward as the average return: $\mathbf{R}_i = Avg(\eta)$;
                     update $i = i + 1$
16              **else if** $h \bmod \tau = 0$ *and* $h \neq H - \tau$ **then**
17                  Construct the reward $\mathbf{R}_i = 0$; update $i = i + 1$

18      Train the hyper-controller **MC** on the data $\{(\mathbf{S}_i, \mathbf{A}_i, \mathbf{R}_i)\}_{i=0}^{m \cdot H / \tau}$
19   **until** *acceptable performance is achieved*

---

Here, $R_i'$ is the average return of MBPO policy trained in advance with the same amount of real data using the original hyperparameters, and we treat $\Sigma_{i=t}^{m \cdot H / \tau} R_i'$ as the baseline. The advantage function can also be regarded as the performance improvement compared with the original MBPO configuration.

## 6   Experiment

In this section, we conduct experiments to verify the effectiveness of AutoMBPO and provide comprehensive empirical analysis. Our experiments aim to answer the following three questions: i) How does the MBPO instance learned by AutoMBPO perform compared to the original configuration? ii) Do the learned hyper-controllers in different target-MDPs share similar schedules, and are the schedules consistent with previous theoretical analysis? iii) Which hyperparameter in MBPO is more important/sensitive to the final performance? More experimental results are provided in Appendix D.

### 6.1   Comparative Evaluation

We mainly compare the MBPO instance found by **AutoMBPO** to the **MBPO** with original hyperparameters [12] since AutoMBPO is directly based on MBPO, and there is no need to compare to other model-based methods. Besides, we also compare to Soft Actor-Critic (SAC) [11], the state-of-the-art model-free algorithm in terms of sample efficiency and asymptotic performance. Actually, SAC can be viewed as an extreme variant of MBPO where the real ratio in training policy is 1. To make a fair comparison, we also provide the result of SAC where the policy training iteration is enforced to be the same as in MBPO. To be more specific, the original SAC, denoted as **SAC(1)**, trains the policy one time per step in the real environment, while in MBPO, the policy is trained 20 times per real step, so we add the baseline **SAC(20)**. We also compare our method to the **PBT** [32] and **RoR** [8] algorithm for a comprehensive comparison. Note that the hyper-controllers in AutoMBPO and RoR are trained in advance and are only evaluated here. The whole training process of AutoMBPO is provided in Appendix D.3.

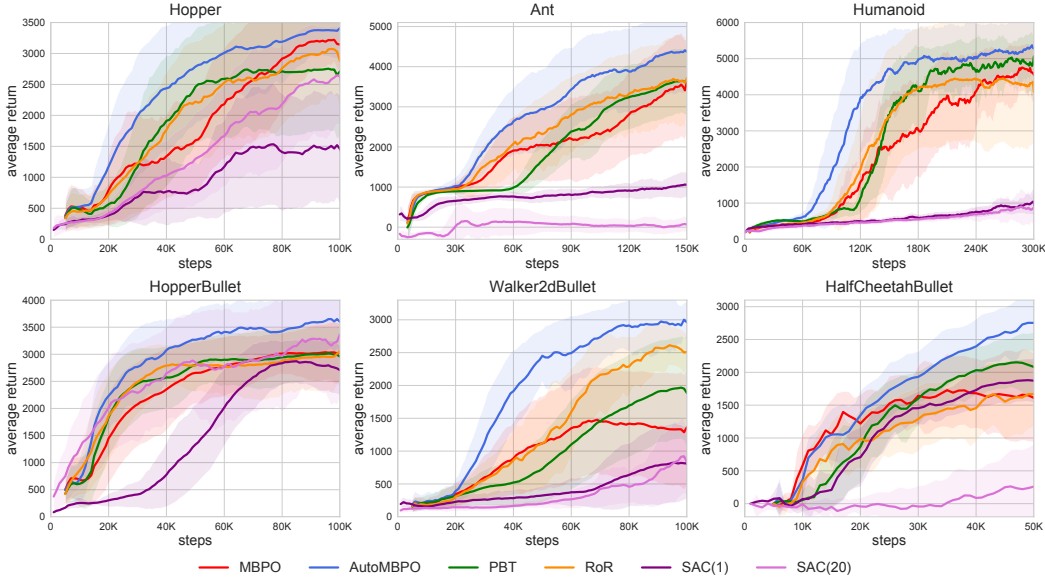

Figure 2: Learning curves of AutoMBPO and baselines on different continuous control environments. The solid lines indicate the mean, and shaded areas indicate the standard deviation of 10 trials over different random seeds. Each trial is evaluated every 1000 environment steps, where each evaluation reports the average return over 10 episodes.

The compared methods are evaluated on three MuJoCo [29] (**Hopper, Ant, Humanoid**), and three PyBullet [6] (**HopperBullet, Walker2dBullet, HalcheetahBullet**) continuous control tasks as our target-MDPs. In practice, the three hyperparameters (real ratio, model training frequency, and policy training iteration) are scheduled on all tasks, while the rollout length is not scheduled on the PyBullet tasks since preliminary experiments showed that even 2-step rollouts degrade the performance on the PyBullet environments. Moreover, due to the high computational cost, the hyper-controller is only trained in previous $m$ episodes of MBPO and will be used to continuously control MBPO instances running for $M$ episodes, which is usually $2 - 3$ times larger than $m$ (e.g., $m = 50, M = 150$ on Ant). Furthermore, we give a penalty for the model training to speed up the training process, and we find it does not influence the results much. More experimental details and hyperparameter configuration for hyper-controller can be found in Appendix F and H.

The comparison results are shown in Figure 2. We can observe that: i) the MBPO instances learned by AutoMBPO significantly outperform MBPO with original configuration in all tasks, highlighting the strength of automatic scheduling. ii) Compared to other hyperparameter optimization baselines (i.e., PBT and RoR), our approach AutoMBPO achieves better performance and generalization in different environments, verifying the effectiveness of our proposed framework. iii) MFRL methods with exhaustive policy training may overfit to a small amount of real data since SAC(20) achieves worse performance than SAC(1) on some complex tasks like Ant and Walker2dBullet. The overfitting problem is much less severe in MBPO since we can generate plentiful data using the model. iv) Though only trained in previous episodes, the hyper-controller can still find reasonable hyperparameter schedules and can be extended to the longer training process of MBPO.

### 6.2 Hyperparameter Schedule Visualization

We visualize the schedules of the hyperparameters found by AutoMBPO in Figure 3. From the results, the hyperparameter schedules on different tasks show both similarities and differences: i) A similar increasing schedule of the real ratio can be observed in all six environments, which is consistent with our derivation in Section 4 and the empirical results in Kalweit and Boedecker [13]. ii) The hyper-controller tends to train the model more frequently in complex environments, such as Ant and Humanoid. iii) To a certain extent, increasing the policy training iteration can better exploit the model. However, it may also increase the risk of instability [22], and we observe that the hyper-controller decreases the policy training iteration in the subsequent training phase of Hopper when the policy

is close to convergence. iv) As for the rollout length, the hyper-controller adopts a nearly linear increasing schedule, which is close to the manually designed one used in the original MBPO [12].

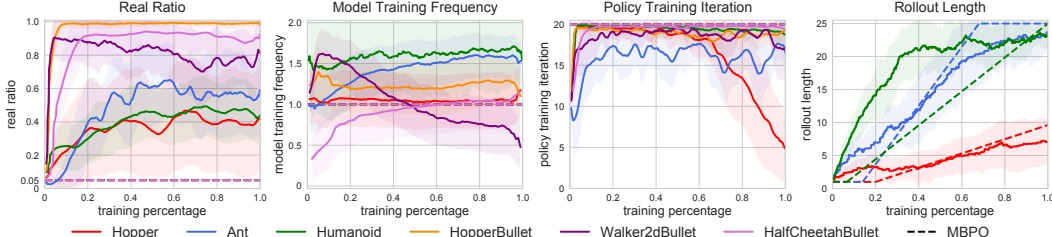

Figure 3: Hyperparameter schedule found by AutoMBPO on different tasks. The dashed lines indicate the original hyperparameter configuration in MBPO. For better visualization, model training frequency is scaled by the frequency of the original MBPO, and the training process (x-axis) is scaled to [0,1] since the max episodes vary on different tasks.

### 6.3 Importance of Hyperparameters

In this section, we investigate the importance of different hyperparameters. Specifically, we utilize the hyper-controller trained previously to schedule one certain hyperparameter respectively, i.e., real ratio (AutoMBPO-R), model training frequency (AutoMBPO-M), policy training iteration (AutoMBPO-P), and rollout length (AutoMBPO-L), while fixing other hyperparameters to the same as in the original MBPO. As shown in Figure 4, the importance of various hyperparameters changes across different tasks: using the hyper-controller to schedule the policy training iteration is helpful on Hopper but fails on Ant, while the situation reverses for model training frequency. However, using the hyper-controller to schedule the real ratio retains much of the benefit of AutoMBPO on almost all tasks, which further verifies the importance of real ratio.

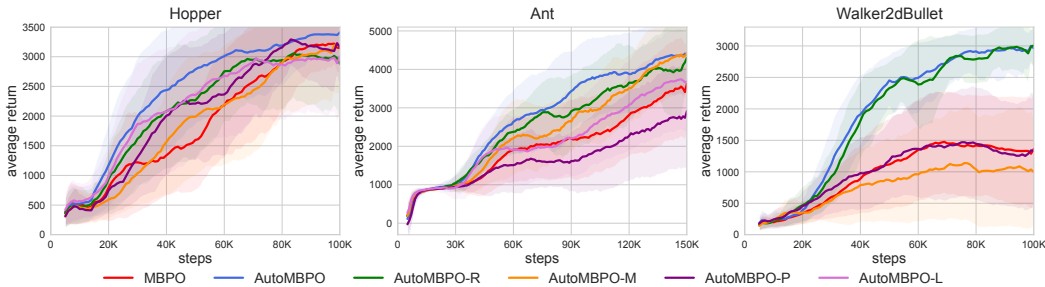

Figure 4: Performance of the instances with only one specific hyperparameter scheduled by hyper-controller while other hyperparameters fixed to the same as in MBPO. The complete figures of all six environments are provided in Appendix D.1.

## 7 Conclusion

In this paper, we conduct a theoretical analysis to justify real data usage for policy optimization in MBRL algorithms. Inspired by the analysis, we present a framework called AutoMBPO to automatically schedule the hyperparameters in the MBPO algorithm. Extensive experiments on complex continuous control tasks demonstrate that the derived MBPO instance with hyperparameters scheduled by AutoMBPO can achieve significantly better performance than the MBPO instance of the original configuration. Moreover, the real ratio schedule discovered by AutoMBPO is consistent with our theoretical analysis. We hope this insight can help design more effective MBRL algorithms. For future work, we plan to apply our hyper-controller training scheme to other MBRL methods.

## Acknowledgments

The SJTU team is supported by "New Generation of AI 2030" Major Project (2018AAA0100900), Shanghai Municipal Science and Technology Major Project (2021SHZDZX0102) and National

Natural Science Foundation of China (62076161, 61772333, 61632017). The work is also sponsored by Huawei Innovation Research Program.

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
