# Appendix for: On Effective Scheduling of Model-based Reinforcement Learning

## A   Proofs

**Assumption A.1.** *(Discounted-average concentrability of future-state distributions) Given $\rho, \mu, m \geq 1$, and an arbitrary sequence of stationary policies $\{\pi_m\}_{m \geq 1}$, assume that the future-state distribution $\rho T^{\pi_1} T^{\pi_2} \ldots T^{\pi_m}$ is absolutely continuous w.r.t. $\mu$. Assume that*

$$c(m) \stackrel{\text{def}}{=} \sup_{\pi_1,\ldots,\pi_m} \left\| \frac{d\left(\rho T^{\pi_1} T^{\pi_2} \ldots T^{\pi_m}\right)}{d\mu} \right\|_{\infty}$$

*satisfies*

$$C_{\rho,\mu} \stackrel{\text{def}}{=} (1-\gamma)^2 \sum_{m \geq 1} m \gamma^{m-1} c(m) < +\infty.$$

We call $c(m)$ the *$m$-step concentrability* of a future-state distribution and call $C_{\rho,\mu}$ the *discounted-average concentrability coefficient* of the future-state distributions. The class of MDPs that satisfies this concentrability assumption is quite large, which is further discussed in Munos and Szepesvári [18].

**Lemma A.1.** *(Pollard, 1984) Let $\mathcal{F}$ be a set of measurable functions $f : \mathcal{X} \to [0, K]$ and let $\epsilon > 0$, $N$ be arbitrary. If $X_i$, $i = 1, \ldots, N$ is an i.i.d. sequence taking values in the space $\mathcal{X}$ then*

$$\mathbb{P}\left( \sup_{f \in \mathcal{F}} \left| \frac{1}{N} \sum_{i=1}^{N} f(X_i) - \mathbb{E}[f(X_1)] \right| > \epsilon \right) \leq 8 \mathbb{E}\left[ \mathcal{N}\left(\epsilon/8, \mathcal{F}\left(X^{1:N}\right)\right) \right] e^{-\frac{N\epsilon^2}{128K^2}},$$

*where $\mathcal{N}_q\left(\epsilon, \mathcal{F}\left(X^{1:N}\right)\right)$ denotes the $(\epsilon,q)$-covering number of the set $\mathcal{F}(x^{1:N}) = \{(f(x_1), \ldots, f(x_N)) \mid f \in \mathcal{F}\}$, i.e., the smallest integer $m$ such that $\mathcal{F}(x^{1:N})$ can be covered by $m$ balls of the normed-space $\left(\mathbb{R}^N, \|\cdot\|_q\right)$ with centers in $\mathcal{F}(x^{1:N})$ and radius $N^{1/q}\epsilon$. And when $q = 1$, $\mathcal{N}$ is used instead of $\mathcal{N}_1$. From the definition, one can esasily see that $\mathcal{N}_q\left(\epsilon, \mathcal{F}\left(X^{1:N}\right)\right) \leq N$.*

**Lemma A.2.** *(Single Iteration Error Bound) Let $V_k$ and $V_{k+1}$ be the value functions of iteration $k$ and $k+1$, and $V_{max} = r_{max}/(1-\gamma)$. For $p \geq 1$ and a certain state distribution $\mu$, let the inherent Bellman error of the value function space $\mathcal{F}$ be defined by $d_{p,\mu}(B\mathcal{F}, \mathcal{F}) = \sup_{V \in \mathcal{F}} \inf_{f \in \mathcal{F}} \|f - BV\|_{p,\mu}$. Assume the value functions are $K_V$-Lipschitz continuous, i.e., for any $k$ and states pair $(s_i, s_j)$, it holds that $|V_k(s_i) - V_k(s_j)| \leq K_V \|s_i - s_j\|_2$. And assume the $L^2$-norm model error between the real next state $s'$ and the predicted ones $\hat{s}'$ obeys a half-normal distribution, i.e., the probability density function $f(\|\hat{s}' - s'\|_2) = \frac{2}{\sqrt{2\pi}\sigma} \exp\left(-\frac{\|\hat{s}' - s'\|_2^2}{2\sigma^2}\right)$. Then for any $\epsilon_0, \delta_0 > 0$, the inequality*

$$\|V_{k+1} - BV_k\|_{p,\mu} \leq d_{p,\mu}(B\mathcal{F}, \mathcal{F}) + \epsilon_0$$

*holds w.p. at least $1 - \delta_0$ provided that*

$$N > 128 \left(\frac{8V_{\max}}{\epsilon_0}\right)^{2p} \left( \log(1/\delta_0) + \log(32N) \right) \tag{10}$$

*and*

$$\sigma < \frac{\epsilon_0}{4\gamma K_V \Phi^{-1}\left(1 - \delta_0/\left(8N|A|(1-\beta)\right)\right)}. \tag{11}$$

*Proof.* Let $f^*$ be the best fit to $BV_k$ in $\mathcal{F}$: $f^* = \Pi_{\mathcal{F}} BV_k$, and let $\hat{\mu}$ denotes the distribution of the $N$ sampled states $\{s_i\}_{i=1}^{N}$. Define $\|\cdot\|_{p,\hat{\mu}}$ as:

$$\|f\|_{p,\hat{\mu}}^p = \frac{1}{N} \sum_{i=1}^{N} |f(s_i)|^p.$$

If the following inequalities hold simultaneously w.p. not smaller than $1 - \delta_0$, then the Lemma can be proved by choosing $\epsilon_0' = \epsilon_0/4$:

$$\|V_{k+1} - BV_k\|_{p,\mu} \leq \|V_{k+1} - BV_k\|_{p,\hat{\mu}} + \epsilon_0' \tag{12}$$

$$\leq \|V_{k+1} - \hat{V}_k\|_{p,\hat{\mu}} + 2\epsilon_0' \tag{13}$$

$$\leq \|f^* - \hat{V}_k\|_{p,\hat{\mu}} + 2\epsilon_0' \tag{14}$$

$$\leq \|f^* - BV_k\|_{p,\hat{\mu}} + 3\epsilon_0' \tag{15}$$

$$\leq \|f^* - BV_k\|_{p,\mu} + 4\epsilon_0' \tag{16}$$

$$\leq d_{p,\mu}(B\mathcal{F}, \mathcal{F}) + 4\epsilon_0'$$

Since $V_{k+1}$ is the best fit to $\hat{V}_k$ in $\mathcal{F}$, (14) holds w.p. 1. So we only need to prove that (12), (13), (15), (16) hold w.p. at least $1 - \delta_0'$ where $\delta_0' = \delta_0/4$. The proof of (12) and (16) is the same as that in (Munos and Szepesvári [18], Lemma1), and we provide a quick proof here for completeness. Let

$$Q = \max\left(\left|\|V_{k+1} - BV_k\|_{p,\mu} - \|V_{k+1} - BV_k\|_{p,\hat{\mu}}\right|, \left|\|f^* - BV_k\|_{p,\mu} - \|f^* - BV_k\|_{p,\hat{\mu}}\right|\right).$$

So (12) and (16) will follow if

$$\mathbb{P}\left(Q > \epsilon_0'\right) \leq \delta_0'. \tag{17}$$

And due to the inequality:

$$Q \leq \sup_{f \in \mathcal{F}}\left|\|f - BV_k\|_{p,\mu} - \|f - BV_k\|_{p,\hat{\mu}}\right|, \tag{18}$$

we have:

$$\mathbb{P}\left(Q > \epsilon_0'\right) \leq \mathbb{P}\left(\sup_{f \in \mathcal{F}}\left|\|f - BV_k\|_{p,\mu} - \|f - BV_k\|_{p,\hat{\mu}}\right| > \epsilon_0'\right). \tag{19}$$

For any event $\omega$ such that

$$\sup_{f \in \mathcal{F}}\left|\|f - BV_k\|_{p,\mu} - \|f - BV_k\|_{p,\hat{\mu}}\right| > \epsilon_0', \tag{20}$$

there exist a function $f' \in \mathcal{F}$ such that

$$\left|\|f' - BV_k\|_{p,\mu} - \|f' - BV_k\|_{p,\hat{\mu}}\right| > \epsilon_0'. \tag{21}$$

First assume that $\|f' - BV_k\|_{p,\hat{\mu}} \leq \|f' - BV_k\|_{p,\mu}$. Hence, $\|f' - BV_k\|_{p,\hat{\mu}} + \epsilon_0' \leq \|f' - BV_k\|_{p,\mu}$. Since the elementary inequality $x^p + y^p \leq (x + y)^p$ holds for $p \leq 1$ and any non-negative x, y, we can get $\|f' - BV_k\|_{p,\hat{\mu}}^p + (\epsilon_0')^p \leq (\|f' - BV_k\|_{p,\hat{\mu}} + \epsilon_0')^p \leq \|f' - BV_k\|_{p,\mu}^p$. And thus

$$\left|\|f' - BV_k\|_{p,\mu}^p - \|f' - BV_k\|_{p,\hat{\mu}}^p\right| > (\epsilon_0')^p. \tag{22}$$

This inequality holds for an analogous reason when $\|f' - BV_k\|_{p,\hat{\mu}} > \|f' - BV_k\|_{p,\mu}$. And since

$$\sup_{f \in \mathcal{F}}\left|\|f - BV_k\|_{p,\mu}^p - \|f - BV_k\|_{p,\hat{\mu}}^p\right| \geq \left|\|f' - BV_k\|_{p,\mu}^p - \|f' - BV_k\|_{p,\hat{\mu}}^p\right|, \tag{23}$$

we can get

$$\mathbb{P}\left(\sup_{f \in \mathcal{F}}\left|\|f - BV_k\|_{p,\mu} - \|f - BV_k\|_{p,\hat{\mu}}\right| > \epsilon_0'\right) \leq \mathbb{P}\left(\sup_{f \in \mathcal{F}}\left|\|f - BV_k\|_{p,\mu}^p - \|f - BV_k\|_{p,\hat{\mu}}^p\right| > (\epsilon_0')^p\right). \tag{24}$$

Observe that $\|f - BV_k\|_{p,\mu}^p = \mathbb{E}\left[|(f(s_1) - BV_k(s_1))|^p\right]$, and $\|f - BV_k\|_{p,\hat{\mu}}^p$ is just the sample average approximation of $\|f - BV_k\|_{p,\mu}^p$. Using Lemma A.1, we can get

$$\mathbb{P}\left(\sup_{f \in \mathcal{F}}\left|\|f - BV_k\|_{p,\mu}^p - \|f - BV_k\|_{p,\hat{\mu}}^p\right| > (\epsilon_0')^p\right) \leq 8\mathbb{E}\left[\mathcal{N}\left(\frac{(\epsilon_0')^p}{8}, \mathcal{F}\left(X^{1:N}\right)\right)\right] e^{-\frac{N}{2}\left(\frac{1}{8}\left(\frac{\epsilon_0'}{2V_{max}}\right)^p\right)^2}. \tag{25}$$

And since the covering number $\mathcal{N}$ satisfies $\mathcal{N}\left((\epsilon_0')^p/8, \mathcal{F}\left(X^{1:N}\right)\right) \le N$, we have

$$\mathbb{P}\left(\sup_{f \in \mathcal{F}}\left|\|f - BV_k\|_{p,\mu}^p - \|f - BV_k\|_{p,\hat{\mu}}^p\right| > (\epsilon_0')^p\right) \le 8Ne^{-\frac{N}{2}\left(\frac{1}{8}\left(\frac{\epsilon_0'}{2V_{max}}\right)^p\right)^2}. \tag{26}$$

Making the right-hand side upper bounded by $\delta_0' = \delta_0/4$ yields a lower bound of $N$, displayed in (10). Then (17) can be proved by combining (19), (24), and (26).

Now we turn to prove (13) and (15). For an arbitrary function $f \in \mathcal{F}$, using the triangle inequality, we have

$$\left|\|f - BV_k\|_{p,\hat{\mu}} - \|f - \hat{V}_k\|_{p,\hat{\mu}}\right| \le \|BV_k - \hat{V}_k\|_{p,\hat{\mu}}. \tag{27}$$

So if we show that $\|BV_k - \hat{V}_k\|_{p,\hat{\mu}} \le \epsilon_0'$ holds w.p. $1 - \delta_0'$, then (13) and (15) can be proved by choosing $f = V_{k+1}$ and $f = f^*$, respectively. To prove $\|BV_k - \hat{V}_k\|_{p,\hat{\mu}} \le \epsilon_0'$, Recall that the data $(s, a, \hat{s}', \hat{r})$ used to calculate $\hat{V}_k$ is sampled from real environment w.p. $\beta$ (case 1), or from the learned model w.p. $1 - \beta$ (case 2).

For case 1, it satisfies that $\hat{s}' = s'$ and $\hat{r} = r$. So we have

$$\mathbb{P}\left(|\hat{r} + \gamma V_k(\hat{s}') - r - \gamma V_k(s')| > \epsilon_0'\right) = 0. \tag{28}$$

And for case 2, like in previous literature [3, 17], assume the reward function is known, i.e., $\hat{r} = r$. We have

$$\begin{aligned}
\mathbb{P}\left(|\hat{r} + \gamma V_k(\hat{s}') - r - \gamma V_k(s')| > \epsilon_0'\right) &= \mathbb{P}\left(\gamma|(V_k(\hat{s}') - V_k(s'))| > \epsilon_0'\right) \\
&\le \mathbb{P}\left(\gamma K_V \|(\hat{s}' - s')\|_2 > \epsilon_0'\right) \\
&= \mathbb{P}\left(\|\hat{s}' - s'\|_2 > \frac{\epsilon_0'}{\gamma K_V}\right).
\end{aligned} \tag{29}$$

By using the model error assumption: $f(\|\hat{s}' - s'\|_2) = \frac{2}{\sqrt{2\pi}\sigma}\exp\left(-\frac{\|\hat{s}' - s'\|_2^2}{2\sigma^2}\right)$, we can write that:

$$\begin{aligned}
\mathbb{P}\left(|\hat{r} + \gamma V_k(\hat{s}') - r - \gamma V_k(s')| > \epsilon_0'\right) &\le \mathbb{P}\left(\|\hat{s}' - s'\|_2 > \frac{\epsilon_0'}{\gamma K_V}\right) \\
&= \mathbb{P}\left(\frac{\|\hat{s}' - s'\|}{\sigma} > \frac{\epsilon_0'}{\gamma K_V \sigma}\right) \\
&= 2\left(1 - \Phi(\frac{\epsilon_0'}{\gamma K_V \sigma})\right),
\end{aligned} \tag{30}$$

where $\Phi$ is the Cumulative Distribution Function (CDF) of the standard normal distribution. Combine these two cases, we can get

$$\mathbb{P}\left(\left|\hat{r} + \gamma V_k(\hat{s}') - r - \gamma V_k(s')\right| > \epsilon_0'\right) \le 2(1 - \beta)\left(1 - \Phi(\frac{\epsilon_0'}{\gamma K_V \sigma})\right). \tag{31}$$

Making the right-hand side upper bounded by $\delta_0'/(N|\mathcal{A}|)$ yields an upper bound of $\sigma$, displayed in (11). And for each sampled state $s_i$, since

$$\left|BV_k(s_i) - \hat{V}_k(s_i)\right| \le \max_{a \in \mathcal{A}}\left|\hat{r} + \gamma V_k(\hat{s}') - r - \gamma V_k(s')\right|,$$

by using a union bounding argument, we can get

$$\mathbb{P}\left(\left|BV_k(s_i) - \hat{V}_k(s_i)\right| > \epsilon_0'\right) \le \delta_0'/N. \tag{32}$$

And by another union bounding argument, we have

$$\mathbb{P}\left(\max_{i=1,...,N}\left|BV_k(s_i) - \hat{V}_k(s_i)\right| > \epsilon_0'\right) \le \delta_0'. \tag{33}$$

And therefore,

$$\mathbb{P}\left( \frac{1}{N}\left| BV_k\left(s_i\right) - \hat{V}_k\left(s_i\right) \right| > \epsilon_0' \right) \leq \delta_0'. \tag{34}$$

Hence, we have proved that $\|BV_k - \hat{V}_k\|_{p,\hat{\mu}} \leq \epsilon_0'$ holds w.p. at least $1 - \delta_0'$. This completes the whole proof. $\qquad\square$

**Lemma A.3.** *(Munos and Szepesvári [18], Theorem 2) Under the concentrability assumption (Assumption A.1), let $\rho$ be an arbitrary state distribution, and $C_{\rho,\mu}$ be the discounted-average concentrability coefficient. If each iteration error can be bounded as $\|V_{k+1} - BV_k\|_{p,\mu} \leq d_{p,\mu}(B\mathcal{F}, \mathcal{F}) + (1-\gamma)^2\epsilon/(4\gamma C_{\rho,\mu}^{1/p})$ w.p. at least $1 - \delta/K$ for $0 \leq k < K$, then the loss due to using $\pi_K$ instead of the optimal policy $\pi^*$ satisfies that w.p. at least $1 - \delta$,*

$$\|V^* - V^{\pi_K}\|_{p,\rho} \leq \frac{2\gamma}{(1-\gamma)^2} C_{\rho,\mu}^{1/p} d_{p,\mu}(B\mathcal{F}, \mathcal{F}) + \epsilon, \tag{35}$$

*provided that*

$$\gamma^K < \left[ \frac{(1-\gamma)^2}{8\gamma V_{\max}}\epsilon \right]^p. \tag{36}$$

Lemma A.3 shows that if the error in each iteration can be bounded with high probability, then the return discrepancy between $\pi_K$ and $\pi^*$ can be bounded with high probability when the number of iterations $K$ is large enough. Moreover, we can set $\rho$ to the distribution of the states we care about more, e.g., the initial states we start to execute $\pi_K$.

**Theorem A.1.** *($\beta$-mixture sampling-based FVI bound) Under the concentrability assumption (Assumption A.1) and the same assumptions of Lemma 4.1, let $\rho$ be an arbitrary state distribution, $C_{\rho,\mu}$ be the discounted-average concentrability coefficient and $\pi_K$ be the greedy policy w.r.t. $V_K$. Let $V^{\pi_K}$ and $V^*$ be the expected return of executing $\pi_K$ and the optimal policy $\pi^*$ in real environment, respectively. Define $N_{\mathrm{real}} = N \cdot |\mathcal{A}| \cdot \beta$ as the expected number of real samples. Then the following bound holds w.p. at least $1 - \delta$:*

$$\begin{aligned}
\|V^* - V^{\pi_K}\|_{p,\rho} \leq & \frac{2\gamma}{(1-\gamma)^2} C_{\rho,\mu}^{1/p} d_{p,\mu}(B\mathcal{F}, \mathcal{F}) + O\left( \left( \frac{\beta|\mathcal{A}|}{N_{\mathrm{real}}}\left( \log(\frac{N_{\mathrm{real}}}{\beta|\mathcal{A}|}) + \log(\frac{K}{\delta}) \right) \right)^{\frac{1}{2p}} \right) \\
& + O\left( \Phi^{-1}\left( 1 - \frac{\beta\delta}{8K N_{\mathrm{real}}(1-\beta)} \right)\sigma \right) + O\left( \gamma^{K/p} V_{\max} \right).
\end{aligned} \tag{37}$$

*Proof.* Lemma A.3 implies that we can bound $\|V^* - V^{\pi_K}\|_{p,\rho}$ w.p. at least $1 - \delta$ as

$$\|V^* - V^{\pi_K}\|_{p,\rho} \leq \frac{2\gamma}{(1-\gamma)^2} C_{\rho,\mu}^{1/p} d_{p,\mu}(B\mathcal{F}, \mathcal{F}) + \epsilon, \tag{38}$$

via bounding $\|V_{k+1} - BV_k\|_{p,\mu}$ w.p. at least $1 - \delta/K$ as

$$\|V_{k+1} - BV_k\|_{p,\mu} \leq d_{p,\mu}(B\mathcal{F}, \mathcal{F}) + (1-\gamma)^2\epsilon/(4\gamma C_{\rho,\mu}^{1/p}). \tag{39}$$

Using Lemma A.2 to bound (39) by setting $\epsilon_0 = (1-\gamma)^2\epsilon/(4\gamma C_{\rho,\mu}^{1/p})$ and $\delta_0 = \delta/K$. We can get the corresponding bounds of $N$ and $\sigma$ w.r.t. $\epsilon$. Combined with (36), we can write $\epsilon$ as

$$\epsilon = O\left( \left( \frac{1}{N}\left( \log(N) + \log(\frac{K}{\delta}) \right) \right)^{\frac{1}{2p}} \right) + O\left( \Phi^{-1}\left( 1 - \frac{\delta}{8KN|\mathcal{A}|(1-\beta)} \right)\sigma \right) + O\left( \gamma^{K/p} V_{\max} \right). \tag{40}$$

Plugging (40) into (38) and substituting $N$ with $N_{\mathrm{real}}/(\beta|\mathcal{A}|)$ complete the proof. $\qquad\square$

## B   Model Error Assumption

In Lemma 4.1 and Theorem 4.1, we assume that the model error between the real next state $s'$ and the predicted ones $\hat{s}'$ obeys a half-normal distribution, i.e., the probability density function $f(\|\hat{s}' - s'\|_2) = \frac{2}{\sqrt{2\pi}\sigma}\exp\left( -\frac{\|\hat{s}' - s'\|_2^2}{2\sigma^2} \right)$. In this section, we give an empirical analysis of this model error assumption. To be more specific, we test the trained dynamics model in MBPO on newly collected data (typically 10k transitions) and plot the frequency of different model error ranges in Figure 5 to approximate the distribution. We can observe that the approximated distribution is close to the half-normal distribution, i.e., the probability of small model error is higher than that of large model error, which empirically shows the rationality of the assumption.

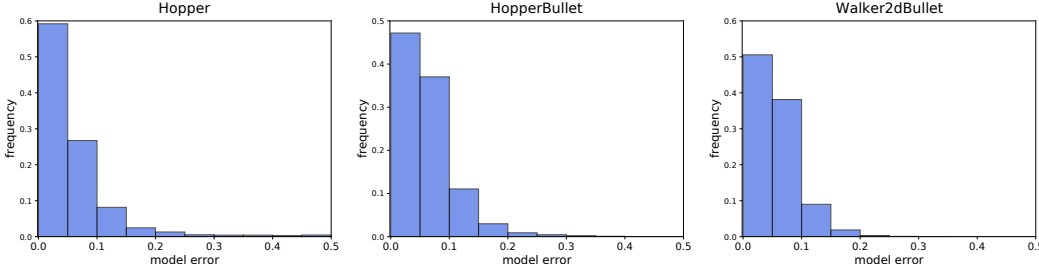

Figure 5: Frequency distribution histogram of the model error $\|\hat{s}' - s'\|_2$. The y-axis indicates the frequency of the corresponding range.

## C    Algorithm Comparison

We compare the hyperparameters that can be scheduled in different Dyna-style MBRL algorithms in Table 1. From the comparison, real ratio and rollout length can only be scheduled in MBPO and MA-BDDPG since other algorithms merely use the imaginary data to train the policy and need to generate model rollouts from the initial state to the end of the episode. Considering that MBPO is much more effective than MA-BDDPG in continuous control benchmark tasks [12, 13], we finally choose MBPO as a representative running case.

Table 1: Comparison of the hyperparameters that can be scheduled in different Dyna-style MBRL algorithms.

|  | Real Ratio | Policy Training Iteration | Model Training Frequency | Rollout Length |
|---|---|---|---|---|
| MBPO [12] | ✓ | ✓ | ✓ | ✓ |
| MA-BDDPG [13] | ✓ | ✓ | ✓ | ✓ |
| SLBO [17] | ✗ | ✓ | ✓ | ✗ |
| ME-TRPO [14] | ✗ | ✓ | ✓ | ✗ |
| MB-MPO [4] | ✗ | ✓ | ✓ | ✗ |
| PAL/MAL [22] | ✗ | ✓ | ✓ | ✗ |

## D    More Experimental Results

### D.1    Hyperparameter Importance

Due to the page limit, we only show the results of hyperparameter importance on three environments in Section 6.3. For a more comprehensive analysis, we plot the results on all the six environments in Figure 6. The conclusion is the same as discussed in Section 6.3, i.e., using the hyper-controller to schedule the real ratio retains much of the advantage of AutoMBPO.

### D.2    Controller Transfer

From the results in Figure 3, we observe that the hyperparameter schedules on the three PyBullet environments are similar, especially for the real ratio $\beta$. Then we further conduct experiments to test whether the hyper-controller trained on these three tasks can transfer to others without additional fine-tuning. Results are shown in Figure 7. Though the performance of the transferred hyper-controller is slightly worse than the original ones, they all surpass the MBPO baseline with a considerable margin, which shows the potential of the hyper-controller to capture the commonality to generalize to different tasks.

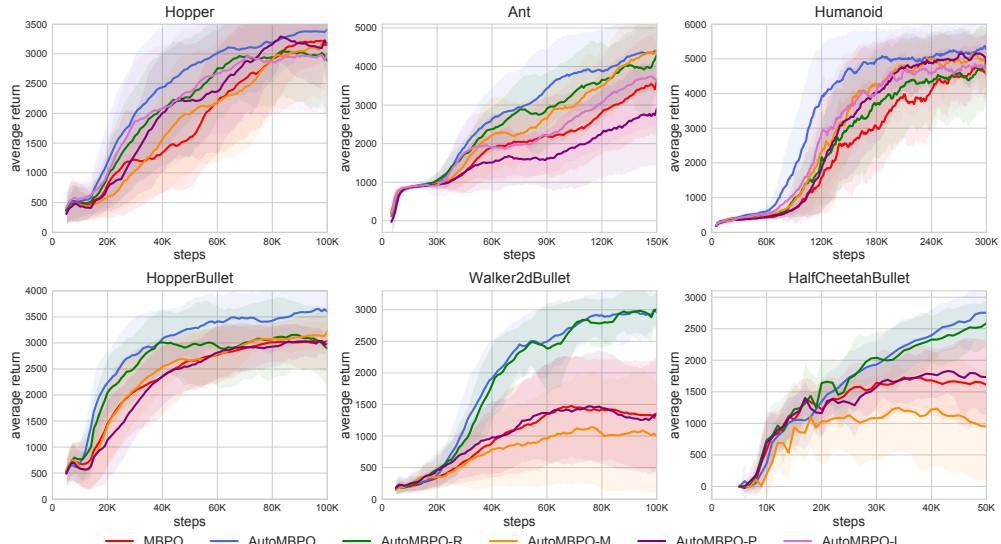

Figure 6: Complete figures of the hyperparameter importance experiments in Section 6.3.

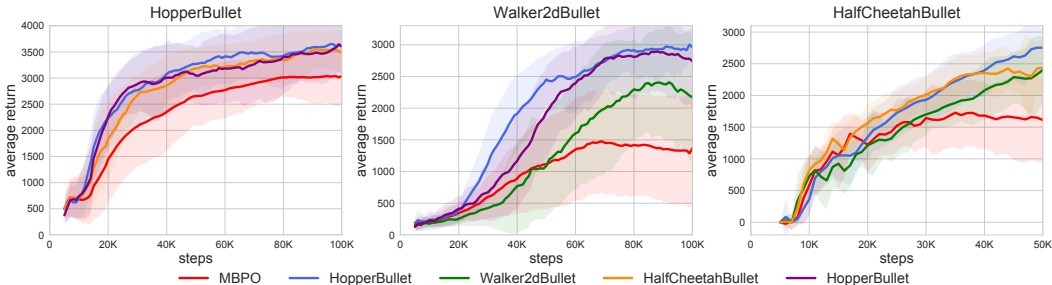

Figure 7: The transfer results of the hyper-controller among PyBullet environments with $A_3^2 = 6$ transferring cases in total. Each figure's title represents the target task, and each specific line indicates the source task. For example, the green line in the middle figure represents transferring the hyper-controller learned on HopperBullet to Walker2dBullet.

## D.3 Effectiveness of AutoMBPO

In Algorithm 1, we train the hyper-controller until it achieves acceptable performance. One concern is whether the hyper-controller improves monotonically during training. We plot the performance of the hyper-controller in different training phases in Figure 8. For example, suppose we train the hyper-controller with 200 hyper-MDP episodes, i.e., 200 MBPO instances in total, then the purple line (1%-20%) represents the average return of the 1-40th MBPO instances, and so on. Monotonic improvement of hyper-controller can be observed during its training phase, which further demonstrates the effectiveness of our algorithm.

Another concern may be the extensive computational requirements for AutoMBPO. According to the computational time table in Appendix G, take Hopper as an example: our algorithm takes about 90 hours to train the hyper-controller, about 5-6 times the time to train a complete MBPO instance, i.e., equivalent to 5-6 trials of MBPO hyperparameters. It is almost impossible to manually find a suitable configuration for these hyperparameters within such few trials.

## D.4 Hyperparameter Study

Since AutoMBPO utilizes the hyper-controller to adjust the hyperparameters of MBPO dynamically, e.g., $\pm 1$, the initial values of these hyperparameters become the hyperparameters of AutoMBPO. We also want to investigate the effect of these initial values on the learning process. In the experiments of Section 6, we initialize the real ratio to $0.05$, the policy training iteration to $10$, and limit the policy training iteration to $[1, 20]$ for computation efficiency. We denote the original one

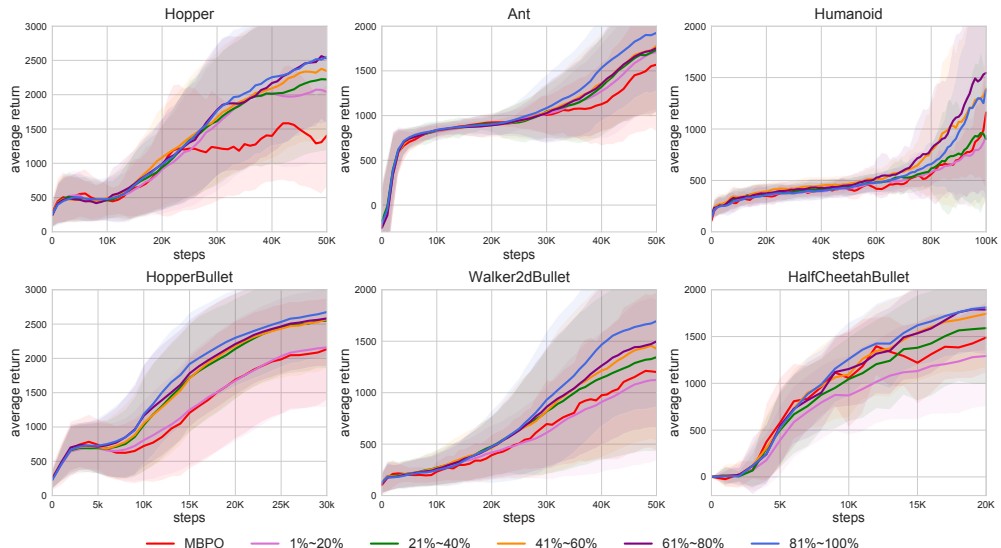

Figure 8: Performance of the hyper-controller in different training phases. Results are the average on six trials of hyper-controller training over different random seeds.

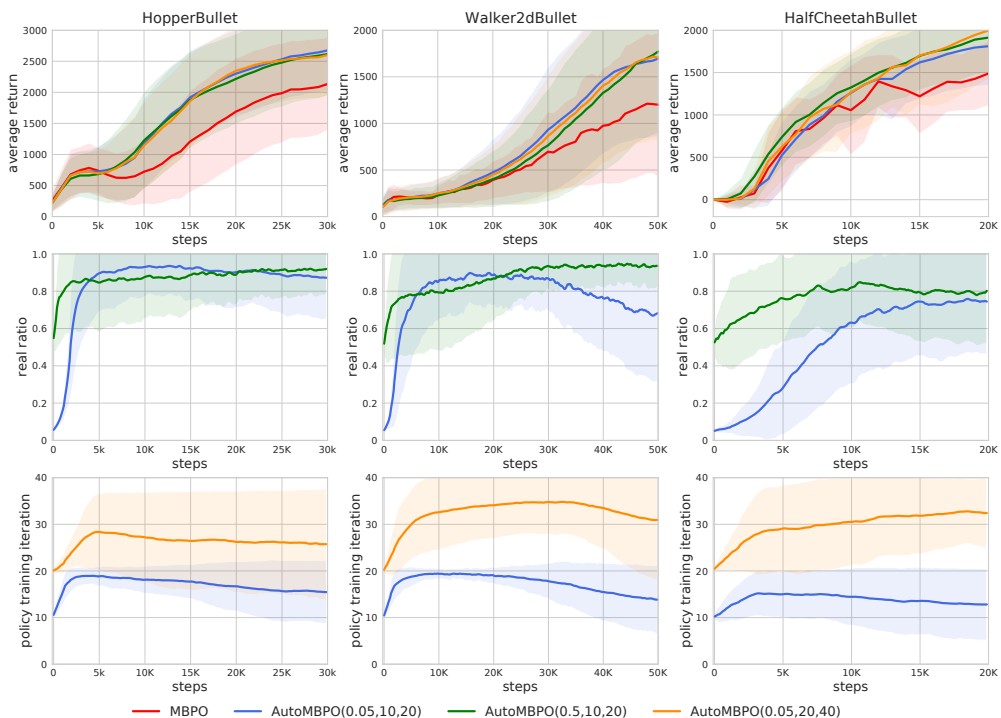

Figure 9: Top: Performance comparison of AutoMBPO(0.5, 10, 20), AutoMBPO(0.05, 20, 40) and the original AutoMBPO(0.05, 10, 20). Middle and bottom: the corresponding schedule of real ratio and policy training iteration.

as AutoMBPO(0.05, 10, 20). In this section, we further conduct experiments of AutoMBPO(0.5, 10, 20) and AutoMBPO(0.05, 20, 40) on three PyBullet environments. The performance and the hyperparameter schedules are shown in Figure 9.

From the comparison, we can find that changing the initial value of real ratio or policy training iteration does not influence the final performance much, which shows the robustness of AutoMBPO to the hyperparameter initialization. Notice that the policy training iteration of AutoMBPO(0.05, 20, 40) is much larger than that of AutoMBPO(0.05, 10, 20), while the performance does not improve

much. This finding is consistent with our conclusion in Section 6.2, i.e., increasing the policy training iteration is not always a good choice.

## D.5 State Feature Ablation

In section 5.1, inspired by Theorem 4.1, we include the number of real samples $N_{\text{real}}$ and the model loss $\mathcal{L}_{\hat{T}}$ into the state formulation of hyper-MDP. In this section, we ablate these two features and only use other features as the state to train the controller, denoted as AutoMBPO-SA. Results are shown in Figure 10. We find that ablating these two features degrades the performance, which highlights the importance of elaborate state design for hyper-controller learning.

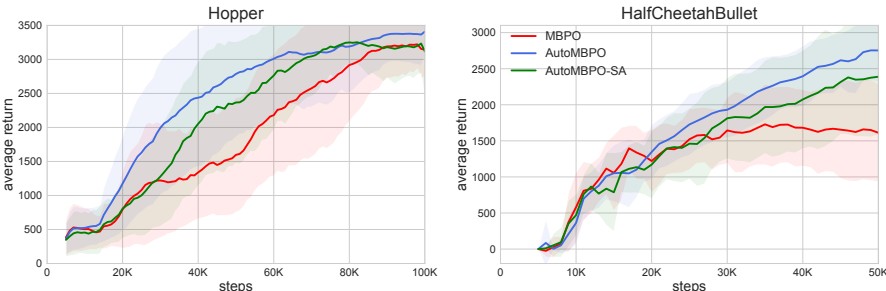

Figure 10: Results of the state feature ablation experiment. AutoMBPO-SA denotes the AutoMBPO variant of excluding real samples number and model loss from the hyper-MDP state.

## D.6 Statistical Significance

Since the shaded areas in Figure 2 overlap, we further add a t-test to the results of the original MBPO and AutoMBPO. We use the average return to perform t-test and list the p-values in Table 2. From the result, the p-values are all less than 0.05, confirming the statistical significance of the results.

Table 2: t-test to the average returns of the original MBPO and AutoMBPO.

|  | Hopper | Ant | Humanoid | Hopper Bullet | Walker2d Bullet | HalfCheetah Bullet |
|---|---|---|---|---|---|---|
| p-value | 0.0078 | 0.0031 | 6e-5 | 0.0052 | 9e-5 | 0.0276 |

# E Baseline Implementation Details

**PBT.** The implementation of PBT mainly follows Zhang et al. [32]. Specifically, we train 10 MBPO instances in parallel, each with randomly initialized hyperparameters. After each episode, 20% of the MBPO instances with low returns are replaced by the top 20% (both hyperparameters, network parameters, and data buffer) or re-initialized (only hyperparameters) with a certain probability.

**RoR.** For RoR, we utilize the same Hyper-MDP definition as AutoMBPO since the original Hyper-MDP definition in Dong et al. [8] is not suitable for MBPO. Moreover, we run RoR 6 times, each using the same amount of data as our method. In the result of Figure 2, we compare to the best performing RoR for a fair comparison.

# F Experimental Settings

We first provide the details of the experimental environments in Table 3. Among them, Hopper, Ant, and Humanoid are the same version used in Janner et al. [12]. For the remaining three PyBullet environments, we modify the reward function to be similar to that of Mujoco to facilitate training since we found that the default reward setting is too complicated for MBPO to solve in preliminary experiments. Then, we present the experimental settings in different environments in Table 4.

Table 3: Environment settings in our experiments. $\theta_t$ denotes the joint angle, $x_t$ denotes the position in x-direction, $a_t$ denotes the action control input, and $z_t$ denotes the height.

| | State Dimension | Action Dimension | Reward Function | Termination States Condition |
|---|---|---|---|---|
| Hopper | 11 | 3 | $\dot{x}_t - 0.001 \|a_t\|_2^2 + 1$ | $z_t \leq 0.7$ or $\theta_t \geq 0.2$ |
| Ant | 27 | 8 | $\dot{x}_t - 0.5 \|a_t\|_2^2 + 1$ | $z_t \leq 0.2$ or $z_t \geq 1.0$ |
| Humanoid | 45 | 17 | $0.25\dot{x}_t - 0.1 \|a_t\|_2^2 + 5$ | $z_t \leq 1.0$ or $z_t \geq 2.0$ |
| Hopper Bullet | 15 | 3 | $5\dot{x}_t - 0.001 \|a_t\|_2^2 + 1$ | $z_t \leq 0.8$ or $|\theta_t| \geq 1.0$ |
| Walker2d Bullet | 22 | 6 | $5\dot{x}_t - 0.001 \|a_t\|_2^2 + 1$ | $z_t \leq 0.8$ or $|\theta_t| \geq 1.0$ |
| HalfCheetah Bullet | 26 | 6 | $5\dot{x}_t - 0.001 \|a_t\|_2^2$ | None |

Table 4: Experimental settings in different environments. Specifically, a hyper-MDP episode consists of m target-MDP episodes, i.e., the whole training process of an MBPO instance, and a target-MDP episode consists of $H$ timesteps in the environments.

| | | Hopper | Ant | Humanoid | Hopper Bullet | Walker2d Bullet | HalfCheetah Bullet |
|---|---|---|---|---|---|---|---|
| | hyper-MDP episodes | 100 | 100 | 200 | 200 | 200 | 100 |
| $m$ | target-MDP episodes for training | 50 | 50 | 100 | 30 | 50 | 20 |
| $M$ | target-MDP episodes for evaluation | 100 | 150 | 300 | 100 | 100 | 50 |
| $H$ | timesteps per target-MDP episode | 1000 | | | | | |

## G Computing Infrastructure

We present the computing infrastructure and the corresponding computational time used to train the hyper-controller in Table 5.

Table 5: Computing infrastructure and the corresponding computational time.

| | Hopper | Ant | Humanoid | Hopper Bullet | Walker2d Bullet | HalfCheetah Bullet |
|---|---|---|---|---|---|---|
| CPU | 32 cores | | | 16 cores | | |
| GPU | RTX2080TI$\times$2 | | | V100 $\times$2 | | |
| computation time in hours | 90.92 | 95.05 | 245.33 | 56.91 | 149.88 | 30.48 |

## H Hyperparameters

Table 6 lists the hyperparameters used in training the hyper-controller. Other hyperparameters of MBPO not scheduled by the hyper-controller are the same as the original one [12]. Note that the hyperparameter $\tau$ in Humanoid varies from that in other environments since the original MBPO

configuration of Humanoid is different. The original MBPO trains the model per 1000 real timesteps in Humanoid but per 250 real timesteps in other environments. So we set $\tau$ to half of the original interval to keep the maximum model training frequency two times of the original configuration.

Table 6: Hyperparameter settings for hyper-controller.

| | | Hopper | Ant | Humanoid | Hopper Bullet | Walker2d Bullet | HalfCheetah Bullet |
|---|---|---|---|---|---|---|---|
| $\tau$ | real timesteps interval per action | 125 | | 500 | 125 | | |
| | policy network architecture | MLP with one hidden layer of size 256 | | | | | |
| | learning rate | $3 \cdot 10^{-4}$ | | | | | |
| | batch size | 64 | | | | | |
| | policy updates per hyper-MDP episode | 30 | | | | | |
| | initial value of real ratio | 0.05 | | | | | |
| | initial value of policy training iteration | 10 | | | | | |
| | initial value of rollout length | 1 | | | | | |
| $\epsilon$ | PPO clip constant | 0.2 | | | | | |
| $c$ | real ratio change constant | 1.2 | | | | | |
| | penalty for each model training | 0.1 | | | | | |