# OpenReview forum: "On Effective Scheduling of Model-based Reinforcement Learning"
_NeurIPS.cc/2021/Conference — NeurIPS 2021 Poster_

### Official Review · Reviewer_yNuW · 2021-07-02

**Rating:** 8
**Confidence:** 4

**Summary:**

The paper is about learning a hyper-parameter setter for some of the hyper-parameters used to train a model-based reinforcement learning algorithm. Theoretical results are provided to show that there exists an optimal ratio of real-data to model-generated data during training. Experiments show that setting hyper-parameters with a parametric policy improves performance over manual hyper-parameter selection or hand-crafted schedules.

**Ethical Concerns:**

I have no ethical concerns.

**Limitations And Societal Impact:**

This is alright.

**Main Review:**

The paper is good at conveying its ideas. The introduction section clearly states what we are going to be dealing with. The related work section does a good job of summarizing the different ways in which models have been used in MBRL.

I am not sure if the theoretical analysis should be placed in such a prominent place, given that it is somewhat dense with notation. Perhaps the results of the analysis can be stated earlier on, or be highlighted more from the rest of this section, such that the first-time reader does not immediately have to confront the math right at the stage when they are most eager to find out about the implementation of the ideas presented in the introduction.

The methods section does a good job of setting up the hyper-MDP. However I have some questions about the execution:

* Algorithm 1 starts with "initialize MBPO". Does that mean the model and policy parameters are re-initialized periodically during the lifetime of the whole algorithm?
* The advantage calculation uses the average return of MBPO trained with the same amount of real data from original parameters. Does this mean using AutoMBPO requires logs from a previous MBPO training run (or a separate process training plain MBPO in parallel and providing such logs)?

The experiments section is well-designed. The comparative results are in favor of the method. The hyper-parameter schedules and the ablation study over the hyper-parameters are both welcome inclusions.

That being said, some points are unclear to me:

* A little confused by the experiments. Does AutoMBPO here mean an experiment where both MBPO and its hyper-controller are being trained in tandem, or has AutoMBPO been trained in advance, and only being applied here? If the two are being trained jointly, shouldn't we see periodic drops in performance, since MBPO gets re-initialized every time a new hyper-episode is started, according to Algorithm 1?
* In case we are indeed training MBPO and its hyper-controller in parallel, I am a little surprised that the scheduling curves in fig. 3 can make sense throughout. I'd expect the hyper-controller to make errors in the beginning of training. Can you comment on this?
* How were the hyper-parameters set for plain MBPO? Is it the result of a hyper-parameter search, or carried over from previous work? If they are carried over, I would be curious about the improvement of AutoMBPO over a near-optimal fixed hyper-parameter setup found by a hyper-parameter search. This is especially true of the bullet-based environments which MBPO was not originally applied to.

All in all, this is a good paper and it would benefit the research community to accept it.

**Other comments**

* Do the theoretical results extend to stochastic transition functions?
* Is it right to refer to the hyper-parameter setting problem as an MDP, given that the state does not fully describe all of the information in the system? I think POMDP would be more accurate here.


**Time Spent Reviewing:**

2

---

> ### Author Response · Authors · 2021-08-10
> **Response to Reviewer yNuW**
>
> Thank you for your valuable comments and suggestions, which are of great help to improve the quality of our work. We will explain your concerns point by point.
>
> **Q1:** "I am not sure if the theoretical analysis should be placed in such a prominent place, given that it is somewhat dense with notation. Perhaps the results of the analysis can be stated earlier on, or be highlighted more from the rest of this section, such that the first-time reader does not immediately have to confront the math right at the stage when they are most eager to find out about the implementation of the ideas presented in the introduction."
>
> **A1:** Thanks for your suggestion. We will revise our paper accordingly in the next version to make it more friendly to readers.
>
> **Q2:** "Algorithm 1 starts with "initialize MBPO". Does that mean the model and policy parameters are re-initialized periodically during the lifetime of the whole algorithm?"
>
> **A2:** Yes, it means that the model and policy of MBPO are re-initialized, including hyper-parameters and network parameters, since the whole training process of an MBPO instance is a hyper-episode. We will make the statement clearer in the next version.
>
> **Q3:** "The advantage calculation uses the average return of MBPO trained with the same amount of real data from original parameters. Does this mean using AutoMBPO requires logs from a previous MBPO training run (or a separate process training plain MBPO in parallel and providing such logs)?"
>
> **A3:** In our implementation, we use the log from a previous MBPO training run. Since we only use the previous m episodes of MBPO to train the hyper-controller,  it does not take much time to get the log of previous m episodes in advance (about 1% of the total training time of the controller). We will make it clearer in the next version.
>
> **Q4:** "A little confused by the experiments. Does AutoMBPO here mean an experiment where both MBPO and its hyper-controller are being trained in tandem, or has AutoMBPO been trained in advance, and only being applied here? If the two are being trained jointly, shouldn't we see periodic drops in performance, since MBPO gets re-initialized every time a new hyper-episode is started, according to Algorithm 1?"
>
> **A4:** In Fig 2 and Fig 3, the hyper-controller of AutoMBPO has been trained in advance and is only applied here. The whole training processes of the hyper-controller are provided in Appendix D.3 due to page limit (Fig 8). As Fig 8 shows, MBPO scheduled by the hyper-controller at the beginning performs similar or even worse than plain MBPO, and its performance gradually increases as the hyper-controller is trained. Such monotonic improvement also reveals the effectiveness of our algorithm. We will provide more experimental details in the next version to make it clearer. For discussion about whether it is worthwhile to train the controller in advance, please refer to A1 to Reviewer AyeP.
>
> **Q5:** "In case we are indeed training MBPO and its hyper-controller in parallel, I am a little surprised that the scheduling curves in fig. 3 can make sense throughout. I'd expect the hyper-controller to make errors in the beginning of training. Can you comment on this?"
>
> **A5:** As described in A4, the hyper-controller is trained in advance, and it does make errors at the beginning of training as it performs worse than plain MBPO in some environments (as Fig 8 shows). However, the hyper-controller soon finds a reasonable schedule and outperforms MBPO after training.
>
> **Q6:** "How were the hyper-parameters set for plain MBPO? Is it the result of a hyper-parameter search, or carried over from previous work? If they are carried over, I would be curious about the improvement of AutoMBPO over a near-optimal fixed hyper-parameter setup found by a hyper-parameter search. This is especially true of the bullet-based environments which MBPO was not originally applied to."
>
> **A6:** For Mujoco-based environments, the hyper-parameters of plain MBPO are the same as the original MBPO paper. And for bullet-based environments, the hyper-parameters are carried over from the original hyper-parameters for Mujoco, and we tuned them slightly for a fair comparison. As the analysis in lines 172-186 shows, the improvement of AutoMBPO over a 'fixed' hyper-parameter setup mainly comes from the advantage of 'dynamic' hyper-parameter schedules.
>
> **Q7:** "Do the theoretical results extend to stochastic transition functions?"
>
> **A7:** The theoretical results can be extended to stochastic transition functions with some modifications. For example, the equations (28)-(31) in Appendix A should be rewritten in the expectation form. We will add a discussion about stochastic transition functions in the next version.
>
> **Q8:** "Is it right to refer to the hyper-parameter setting problem as an MDP, given that the state does not fully describe all of the information in the system? I think POMDP would be more accurate here."
>
> **A8:** POMDP may be more accurate here since the true state should contain all the information in the system, including all network parameters and collected data of MBPO. However, it is too much for the controller to process. Therefore, we select the critical training information which we think will help the hyper-controller make decisions. We are happy to provide more discussions in the next version.

---

> > ### Comment · Reviewer_yNuW · 2021-08-12
> > **Thank you**
> >
> > I thank the authors for providing an extensive clarification. I've also read their responses to the other reviewers. I can see that the main concern here is about the sample- and computational-complexity of the approach, which is indeed quite high ("5-6 times that of regular MBPO" according to an author comment). Still, given the scope of the work and the difficulty of tuning model-based RL algorithms, I still think that this is a good paper and stand by my original rating.

---

### Official Review · Reviewer_AyeP · 2021-07-09

**Rating:** 6
**Confidence:** 3

**Summary:**

This work seeks to bring some clarity the hyperparameter management of Dyna-style deep RL (i.e. MBPO).
The main parameter of interest being the real data to synthetic data ratio.
It does this through theoretical analysis which is followed up by an automl framework in the style of 'RL for RL' (RoR).
Training an outer policy to schedule policy performs better than standard MBPO across several tasks.

**Limitations And Societal Impact:**

No issues

**Main Review:**

The MBPO algorithm and DYNA-style RL algorithms are very popular at the moment, so I can see this work being of interest to the deep RL community.
The outer policy does seem to have a widespread improvement on standard MBPO in the mean and the theoretical analysis looks correct.

I see this paper more as an analysis into hyperparameters rather than a practical MBRL algorithm.
The need to train a PPO outer loop to me throws away all the sample efficiency that MBRL is supposed to provide, and judging form the hyperparameters in the appendix the full algorithm requires ~1e6 samples that model-free RL usually requires.
I would be useful to see the 'full' training curves (including the outer loop) to see who effective the outer RL is.

The paper often refers to this method as 'simple yet effective'. I would disagree that a bilevel deep RL problem is 'simple'.
The phrase 'simple yet effective' also a suggests that its a surprise that this method works, whereas for me it is more a question of whether the extensive computational requirements are worthwhile for the task in hand.

It is also discussed that the empirical results verify that the theory is correct and that it is optimal to slowly increase the data percentage from low to high. However, in the experiements it looks like the initial value of the ratio is set to 0.05 (Table 4), so the policy appears biased to start low.  I think a more uniform initial exploration is required before that claim can be made.

A rather specific observation I would like some clarification on: on HopperBullet the ratio quickly goes to 1.0, which suggests that MBPO should behave the same as SAC(20) within 10% of training (Fig 3), yet in the performance (Fig 2) AutoMBPO is worse that SAC(20) for the first ~10% but then is better for the remaining 90%. Shouldn't the performance be the same?

Given the amount of computation required to run this bi-level optimizaiton, it seems the performance increase is not really statistically significant apart from on WalkerBullet. I think some discussion of the statistical significance of the results (since requires so much more computation) would benefit the reader.

Regarding the baselines, I would appreciate more of a discussion in the appendix regarding how PBT and RoR are implemented as baselines.

Bayesian optimization is a popular method for HPO. I wonder if, rather that optimizing a neural network policy, you could define a 'schedule spline' (or some similar parameterization) that is only a few parameters but covers a sensible space of possible schedules.

Minor
line 4 of algorithm 1, typo on target

**Time Spent Reviewing:**

3

---

> ### Author Response · Authors · 2021-08-10
> **Response to Reviewer AyeP**
>
> Thanks for the time and effort you have spent reviewing our paper. We will explain your concerns point by point.
>
> **Q1:** "I see this paper more as an analysis into hyper-parameters rather than a practical MBRL algorithm. The need to train a PPO outer loop to me throws away all the sample efficiency that MBRL is supposed to provide, and judging form the hyper-parameters in the appendix the full algorithm requires ~1e6 samples that model-free RL usually requires. I would be useful to see the 'full' training curves (including the outer loop) to see who effective the outer RL is."
>
> **A1:** We have provided the 'full' training curves in Appendix D.3. Monotonic improvement of the outer RL can be observed from the result (Fig 8). That is, the performance of AutoMBPO becomes better as the hyper-controller is trained, which shows the effectiveness of the outer RL.
>
> You are right that we are not meant to propose a practical MBRL algorithm to have minimal total sample complexity during the whole controller training. Instead, we aim to investigate how to schedule the hyperparameters to achieve better performance of Dyna-style MBRL methods, and we hope the hints we conclude can help design potentially better algorithms.
>
> Besides, it is worth mentioning that though the training of outer PPO may sacrifice some sample efficiency, it can still reduce the effort and time to tune the MBRL hyper-parameters manually. Take Hopper as an example: our algorithm takes about 90 hours to train the controller, about 5-6 times the time to train a complete MBPO instance, i.e., equivalent to 5-6 trials of MBPO hyperparameters.  It is almost impossible to manually find a suitable configuration for these hyper-parameters within such few trials. We will provide more discussion about the effectiveness of the outer RL in the next version.
>
> **Q2:** "The paper often refers to this method as 'simple yet effective'. I would disagree that a bilevel deep RL problem is 'simple'. The phrase 'simple yet effective' also a suggests that its a surprise that this method works."
>
> **A2:** We call the method 'simple' mainly because that the outer RL is just a vanilla PPO (2 hidden layers for policy network, without additional modification), and the hyper-parameters for the outer RL in different environments are almost the same and easy to determine, as listed and discussed in Appendix F. The statement 'simple yet effective' is inaccurate and leads to ambiguity. We will fix this expression problem in the next version.
>
> **Q3:** "Whereas for me it is more a question of whether the extensive computational requirements are worthwhile for the task in hand."
>
> **A3:** As discussed in A1, our algorithm takes about 5-6 times the time to train a complete MBPO instance. Considering the complexity of MBRL methods and the interplay between different hyper-parameters, we think it is worthwhile since it is almost impossible to manually find a suitable configuration for several hyper-parameters within such few runnings, especially when the optimal hyper-parameters setting is dynamic.
>
> **Q4:** "It is also discussed that the empirical results verify that the theory is correct and that it is optimal to slowly increase the data percentage from low to high. However, in the experiements it looks like the initial value of the ratio is set to 0.05 (Table 4), so the policy appears biased to start low. I think a more uniform initial exploration is required before that claim can be made."
>
> **A4:** We have provided a study of the initial value of the ratio in Appendix D.4. Specifically, we conducted experiments with the initial value of the ratio set to 0.5 (Fig 9), which also learns an increasing ratio schedule similar to the results of 0.05. We will provide a more detailed discussion about the initial value in the next version.
>
> **Q5:** "A rather specific observation I would like some clarification on: on HopperBullet the ratio quickly goes to 1.0, which suggests that MBPO should behave the same as SAC(20) within 10% of training (Fig 3), yet in the performance (Fig 2) AutoMBPO is worse that SAC(20) for the first ~10% but then is better for the remaining 90%. Shouldn't the performance be the same?"
>
> **A5:** Though the real ratio quickly goes to 1.0, it still uses some model-generated data in the first 10% of training. Besides, the difference in policy training iteration (Fig 3) may also affect the performance: AutoMBPO increases policy training iteration from 10 to about 18, while SAC(20) fix it to 20.  In other words, SAC(20) updates the policy too frequently in the first 10% of training using only the real data, which may make it improve fast at the beginning but easier converge to a local optimum.
>
> **Q6:** "Given the amount of computation required to run this bi-level optimizaiton, it seems the performance increase is not really statistically significant apart from on WalkerBullet. I think some discussion of the statistical significance of the results (since requires so much more computation) would benefit the reader."
>
> **A6:** As discussed in A1, the computational cost is not so much compared with the cost of manually tuning the hyper-parameters of an MBRL algorithm. Besides, apart from on WalkerBullet, the performance improvements in other environments are also considerable. For example, on Hopper and Ant, AutoMBPO reduces the real data used to reach 3000 average return by about 30%, and on HopperBullet and HalfCheetahBullet, the asymptotic performance of AutoMBPO is about 600-1000 higher than that of MBPO. Moreover, we have run 10 trials over different random seeds for the result in Fig 2 (5 trials in the original MBPO paper) to make the result more convincing statistically. We further add a t-test to the results of the original MBPO and AutoMBPO. We use the average return during the whole training process to perform t-test and list the p-values below:
>
> ||Pendulum|Hopper|Ant|HopperBullet|WalkerBullet|HalfCheetahBullet|
> | :------: | :------: | :------: | :------: | :------: | :------: | :------: |
> |p-value|0.0169|0.0078|0.0031|0.0052|9e-5|0.0276|
>
> The p-values are all less than 0.05, confirming the statistical significance of the results. We will provide more discussion of the statistical significance of the results in the next version.
>
> **Q7:** "Regarding the baselines, I would appreciate more of a discussion in the appendix regarding how PBT and RoR are implemented as baselines."
>
> **A7:** We will provide more discussion about the implementation details of baselines in the appendix. Specifically, we mainly follow the implementation of [1] and [2]. For PBT, we train 10 MBPO instances in parallel, and after each episode, 20% of the MBPO instances with low returns are replaced by the top 20% (both hyper-parameters and network parameters). And for RoR, we run RoR 6 times, each time using the same amount of data as our method. In the result as Fig 2 shows, we compare to the best performing RoR for a fair comparison.
>
> **Q8:** "Bayesian optimization is a popular method for HPO. I wonder if, rather that optimizing a neural network policy, you could define a 'schedule spline' (or some similar parameterization) that is only a few parameters but covers a sensible space of possible schedules."
>
> **A8:** As discussed in [1], Bayesian optimization may not be suitable for MBRL due to the complexity of MBRL algorithms. It is interesting to define a 'schedule spline' covering a sensible space of possible schedules. However, the 'schedule spline' definition can be a big challenge, requiring much manual effort, and the schedule space is still limited. We are happy to give it a try in the future.
>
> **Q9:** "Minor line 4 of algorithm 1, typo on target."
>
> **A9:** Thank you for your comments. We will fix this typo in the next version.
>
> [1] Zhang, Baohe, et al. "On the importance of hyper-parameter optimization for model-based reinforcement learning." International Conference on Artificial Intelligence and Statistics. PMLR, 2021.
>
> [2] Dong, Linsen, et al. "Intelligent Trainer for Dyna-Style Model-Based Deep Reinforcement Learning." IEEE Transactions on Neural Networks and Learning Systems (2020).

---

> > ### Comment · Reviewer_AyeP · 2021-08-26
> > **Response**
> >
> > Thanks for this, I'm generally happy.
> >
> > Re: A3, what about finding hyperparams for the outer look? The logic is a bit flimsy.

---

> > > ### Author Response · Authors · 2021-08-26
> > > **New response**
> > >
> > > Thanks for your reply. The hyper-parameter of the outer loop is also one concern of Reviewer SPME, and we have answered the question (see A3) in the first response to him/her.
> > >
> > > We claim finding the hyper-parameter of the outer loop did not take much effort for the following two reasons:
> > > 1. According to the hyper-parameter study in Appendix D.4, our algorithm performs similarly with different hyper-parameters, which shows that it is not sensitive to these hyper-parameters.
> > > 2. According to Appendix F, we use nearly the same hyper-parameters in all the environments, and the results are all effective. Moreover, when we conducted the experiments on Humanoid as Reviewer SPME suggested, we also used the same hyper-parameter, and our algorithm still performs well (see results in the second response to Reviewer SPME).

---

> ### Author Response · Authors · 2021-08-15
> **Thanks for your comment. We are willing to address further concerns.**
>
> We appreciate your valuable and constructive comments, which provide much helpful guidance to improve the quality of our paper. We hope our last reply has resolved all your concerns. If you have any other questions, we are also pleased to respond. We sincerely look forward to your response.

---

> ### Comment · Area_Chair_auRo · 2021-08-23
> **Did the author addressed your concerns?**
>
> Dear reviewer, did the author addressed your concerns? Thanks! --Your AC

---

### Official Review · Reviewer_SPME · 2021-07-16

**Rating:** 6
**Confidence:** 3

**Summary:**

The paper deals with the problem of hyper-parameters optimization in model-based reinforcement learning (MBRL) Specifically, the paper tries to answer the question of how to schedule hyperparameters such as real data ratio, model training frequency, policy training iteration, and rollout length to achieve better performance. Supplementals materials are provided.

There are 2 contributions in the paper:
- The paper provides some analysis to show that real data ratios affect the performance of MBRL. Specifically, they show that increasing the real data ratio improves the performance of MBRL.
- They propose a new MBRL algorithm that learns to schedule hyper-parameters on the fly.


**Limitations And Societal Impact:**

The authors did not provide limitations and societal impact.

**Main Review:**

Main Strengths:
- The paper makes an effort to analyze the effect of real data ratio on the performance of MBRL. The analysis is based on the one by Munos and Szepesvári but the author did a good job in modifying the proof in the mixed data setting.
- The author proposes a hyper-parameters tuning method based on Reinforcement Learning. Although the idea is not new, the author did a good job by choosing the appropriate MDP setup.
- The experiments show good results of their proposed method.
- Overall, the paper is well written. The way the authors present the paper is easy to follow.

Main Weaknesses:
- There is one problem with the comment that the author made about their bound. Since Nreal = N · |A| · β, the ratio β/Nreal  is a constant. Thus it is not clear how gradually increasing the real ratio β is promising to achieve good performance according to the upper bound as what the authors claim.
- The proposed algorithm and the theoretical analysis do not have any relation. The analysis only shows that the real data ratio plays an important role in MBRL. Besides, the theory does not provide any hints in the design of the proposed algorithm. However, the authors claim that their algorithm is inspired by the theory, which is not totally correct in my humble opinion.
- The idea of using RL to tune hyper-parameters is not new and so not a novel contribution. Intuitively, the paper might get trapped in a no-ending route. In detail, the purpose of the paper is to improve MBRL by dynamically tuning hyper-parameters. However, the authors again propose another RL algorithm to achieve this purpose. How are you going to efficiently tune the hyper-parameters of this new RL algorithm? Please let me know if I miss something here.

Question: Have you tried your model on Humanoid and if so what was the result?


**Time Spent Reviewing:**

10

---

> ### Author Response · Authors · 2021-08-10
> **Response to Reviewer SPME**
>
> Thanks for your valuable and constructive comments. We will explain your concerns point by point.
>
> **Q1:** "Since $N_{\rm{real}} = N \cdot |\mathcal{A}|\cdot \beta$, the ratio $\beta / N_{\rm{real}}$ is a constant. Thus it is not clear how gradually increasing the real ratio $\beta$ is promising to achieve good performance according to the upper bound as what the authors claim."
>
> **A1:** We have $\beta / N_{\rm{real}} = 1/(N · |A|)$, where $N· |A| =N_{\rm{real}} + N_{\rm{fake}}$ denotes the amount of all data used for policy optimization.  Since we can generate any amount of fake data using the model, $N_{\rm{fake}}$ is not a constant, nor is $\beta / N_{\rm{real}}$. So we can gradually reduce the amount of fake data to increase the real ratio $\beta$, which is promising to achieve good performance according to the upper bound, as discussed in lines 172-180. We will make the statement clearer in the next version.
>
> **Q2:** "The proposed algorithm and the theoretical analysis do not have any relation. The analysis only shows that the real data ratio plays an important role in MBRL. Besides, the theory does not provide any hints in the design of the proposed algorithm. However, the authors claim that their algorithm is inspired by the theory, which is not totally correct in my humble opinion."
>
> **A2:** The theory shows that there exists the optimal real ratio schedule. However, it is not trivial to determine the specific schedule in practice. So it motivates us the idea of automatically scheduling the real ratio, and we use RL to implement it. Besides, the analysis inspires us to design our method in the following three aspects.
> * Firstly, the theory motivates the design of the Hyper-MDP state. The analysis in lines 188-190 suggests adjusting the real ratio depends on the current real data number, model error, and real ratio used currently. So we include all the information in the Hyper-MDP state to help the controller make decisions.
> * Secondly, the theory provides hints for the hyper-controller to schedule different hyper-parameters. As discussed in lines 190-192, the  Hyper-MDP state may be affected by other hyper-parameters, e.g., model training frequency and rollout length, which reveals the necessity to schedule real ratio and other hyper-parameters jointly, rather than scheduling them individually.
> * More importantly, the theory also guides the design of Hyper-MDP action space (i.e., multiplying a constant c).  An alternative way is that the controller directly outputs the value of real ratio, which has a much larger search space and will bring more difficulty for controller training. According to the analysis in lines 176-180, there exists an optimal value for $\beta / N_{\rm{real}}$. Together with the fact that $N_{\rm{real}}$ increases slowly during training, we are inspired that adjusting the real ratio $\beta$ by multiplying a constant c is enough to get an effective schedule.
>
> We will provide a more detailed discussion about the relation between the proposed algorithm and the theoretical analysis in the next version.
>
> **Q3:** "The idea of using RL to tune hyper-parameters is not new and so not a novel contribution. Intuitively, the paper might get trapped in a no-ending route. In detail, the purpose of the paper is to improve MBRL by dynamically tuning hyper-parameters. However, the authors again propose another RL algorithm to achieve this purpose. How are you going to efficiently tune the hyper-parameters of this new RL algorithm? Please let me know if I miss something here."
>
> **A3:** The main contributions of this work are 1). the theoretical analysis of the real ratio schedule; 2). the hyper-MDP design motivated by the theory; 3). the insights obtained from the empirical results. In Appendix F, we have listed all the hyper-parameters of the new RL algorithm (controller) and discussed why we choose them. In short, It did not take much effort to tune them, and we use nearly the same hyper-parameters in all the different environments. Besides, according to the hyper-parameter study in Appendix D.4, we find the new RL algorithm performs similarly with different hyper-parameters, which shows that the new RL algorithm is not sensitive to these hyper-parameters.
>
> **Q4:** "Have you tried your model on Humanoid and if so what was the result?"
>
> **A4:** We did not try the Humanoid environment in the first version considering the following reasons:
> * The Humanoid environment requires much more computational resources than Hopper or Ant since the original MBPO utilizes a larger neural network for Humanoid (twice hidden units of other environments).
> * As described in lines 288-291, the controller is only trained in previous m episodes of MBPO to reduce the computational cost, e.g., m = 50 in Ant. However, in the result curve of the original MBPO paper, MBPO does not achieve much improvement in the previous 50 episodes on Humanoid. This means we need to use larger MBPO training episodes m to train the controller (e.g., m = 100), further lifting the computational cost.
>
> We tried the Humanoid environment after we read your comment, but the experiments have not finished yet. From current logs, AutoMBPO outperforms MBPO by about 20% in the previous episodes. We report the average return at 100 episodes  of MBPO and AutoMBPO below:
>
> ||MBPO|AutoMBPO|
> |:------:|:------:|:------:|
> |average return (100 episodes)|1060.9|1316.8|
>
> And we will add the complete result on Humanoid in the next version.

---

> ### Author Response · Authors · 2021-08-15
> **Thanks for your comment. We are willing to address further concerns.**
>
> We appreciate your valuable and constructive comments, which provide much helpful guidance to improve the quality of our paper. We hope our last reply has resolved all your concerns. If you have any other questions, we are also pleased to respond.  We sincerely look forward to your response.
>
> Moreover, the experiment on Humanoid has finished now. The hyper-controller on Humanoid also learned an increasing real ratio schedule, and we report the average return at different episodes below:
>
> ||50 episodes|100 episodes|150 episodes|200 episodes|250 episodes|300 episodes|
> |:------:|:------:|:------:|:------:|:------:|:------:|:------:|
> |MBPO          |332.2|1060.9|2125.5|3434.9|4474.4|4550.1|
> |AutoMBPO|469.8|1316.8|2745.4|4702.8|4950.2|5200.5|
>
> By the way, the experiment on Humanoid utilizes the same hyper-parameters as other environments, which further suggests that it does not take much effort to determine the hyper-parameters of the new RL algorithm (as discussed in A3 of the last reply).

---

> ### Author Response · Authors · 2021-08-20
> **We sincerely look forward to your reply.**
>
> Dear reviewer,
>
> We first thank you again for your valuable comments and suggestions. In the previous replies, we think we have addressed your questions point by point and added the experiments on Humanoid as you suggested. We sincerely look forward to your reply to our response.
>
> Best wishes!
> The authors.

---

### Decision · Program_Chairs · 2021-09-27

**Decision:**

Accept (Poster)

**Comment:**

Reviewers agree that the paper is well-motivated, well-written and the the proposed method, motivated by theoretical analysis, shows convincing experimental results on hyperparameter tuning of model-based RL, a hard yet important problem to address. On the other hand, minor concerns remain mainly about whether the theory is connected well with the practice, whether another outside RL loop over model-based RL is "too heavy" in practice and requires high sample complexity.  Authors have address many of the concerns raised by the reviewers and I am happy to accept this paper.